# GenFlow: Constrained Long-Form Text Generation via Adaptive Workflow Optimization

## Abstract

Large Language Models (LLMs) exhibit strong abilities in generating coherent human-like text, yet producing long-form content that satisfies complex constraints remains challenging. Existing approaches either extend generation length through large curated datasets, as in LongWriter, or structure outputs via cognitive-inspired hierarchical planning, as in CogWriter, but often struggle to balance coherence, semantic fidelity, and explicit requirements. In this work, we propose GenFlow, an adaptive framework for constrained long-form text generation. It decomposes writing objectives into constraint-aware sub-plans, uses adaptive decision-making and reward filtering to retain high-quality plans, and optimizes both local and global generation. By embedding constraints directly into the workflow, GenFlow ensures consistency while adapting to evolving requirements. Experimental results on the Qwen2.5 series demonstrate that GenFlow outperforms GPT-4o-mini and CogWriter baselines in constraint satisfaction, coherence, and overall quality. Our code is publicly available[1].

## 1 Introduction

Large Language Models (LLMs) (Liu & et al., 2023), such as GPT (Brown & et al., 2020), achieve human-like fluency and coherence, advancing automated long-form content creation (Bai & et al., 2024; Wan & et al., 2025). However, generating extended text that satisfies complex constraints remains challenging (Pham & et al., 2024; Menchaca Resendiz & Klinger, 2024), as it requires optimizing multiple objectives while adapting to evolving requirements. As illustrated in Figure 1, constrained long-form text generation requires understanding the user's objectives, determining the structural and content requirements they involve, and organizing the production of different parts of the text so that the final document remains coherent and satisfies all constraints. This task forms a workflow in which each step must work together and adapt to evolving requirements, with feedback signals guiding the overall process (Li & et al., 2024).

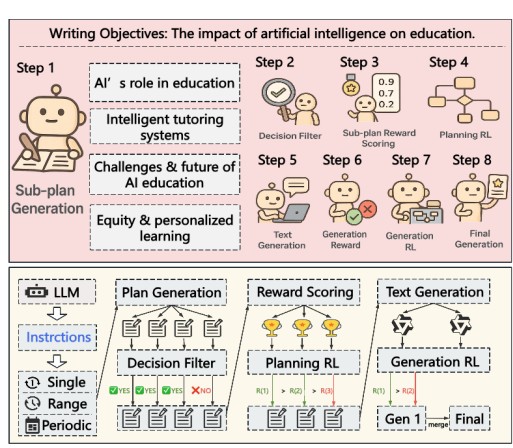

Figure 1: An illustration of GenFlow framework.

Despite recent progress in long-form text generation, existing approaches still struggle to balance structural coherence, semantic fidelity, and constraint satisfaction. To address these challenges, prior studies have followed two complementary directions. Data-centric approaches, exemplified by *LongWriter* (Bai & et al., 2024), extend generation length by constructing synthetic long-form datasets and applying preference optimization inspired by InstructGPT-style RLHF (Ouyang et al., 2022). While effective in scaling output length, their performance depends heavily on large curated datasets and lacks adaptability to evolving constraints (Gilardi et al., 2023). Cognitive-inspired frameworks, such as *CogWriter* (Wan & et al., 2025), builds on Cognitive Writing Theory (Flower & Hayes, 1981), which views human writing as a recursive process of planning, translating, reviewing,

---

[1]Anonymous Github Link: `https://anonymous.4open.science/r/GenFlow`

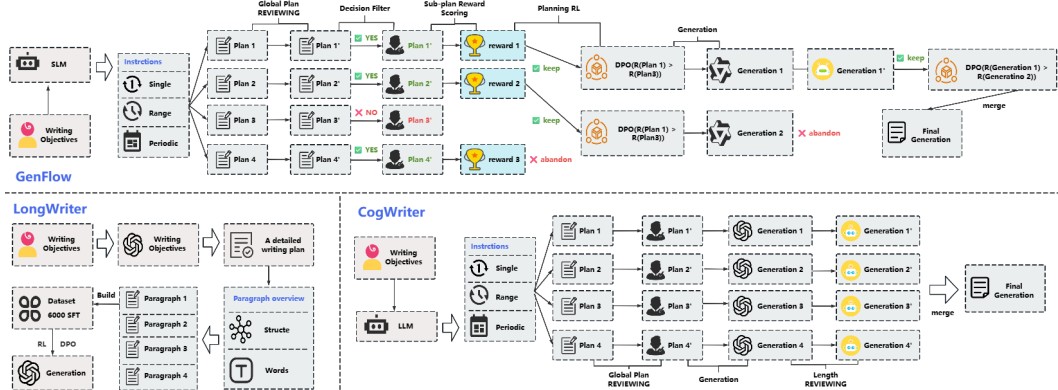

Figure 2: Framework comparison of GenFlow, LongWriter, and CogWriter, highlighting key differences in planning, reviewing, and generation for producing coherent, high-quality long-form text.

and monitoring. By structuring generation through hierarchical planning and multiple agents (Yao et al., 2023b), these methods improve organization and coherence but incur high computational overhead (Zhang et al., 2022a) and offer limited mechanisms for explicit reward integration.

Despite these advances, several challenges remain: **(i) Constraint-Aware Workflow Design:** Constrained long-form generation requires balancing structure, semantics, and explicit requirements, yet existing workflows are linear and fail to embed constraints throughout. **(ii) Adaptive Workflow Optimization:** Current methods rely on static datasets or costly refinement, making them inflexible to evolving requirements; a more dynamic optimization mechanism is needed. **(iii) Feedback-Coupled Coordination:** Planning, generation, and alignment are often decoupled, with feedback applied post hoc, limiting real-time guidance of local and global decisions.

To address these challenges, we propose GenFlow, an Adaptive Workflow Optimization framework that integrates constraint satisfaction with dynamic workflow design. GenFlow treats long-form generation as an adaptive workflow where planning, decision-making, generation, and feedback interact in a unified loop. It decomposes objectives into constraint-aware sub-plans via hierarchical planning, applies decision and reward-based filtering to retain high-quality candidates, and guides both local generation and global assembly. By embedding constraint signals into each stage, GenFlow ensures coherence, enforces structural and stylistic requirements, and adapts to evolving user demands. As shown in Figure 2, GenFlow differs from CogWriter (Wan & et al., 2025) and LongWriter (Bai & et al., 2024) through its workflow-centric design and adaptive optimization strategy.

We evaluate GenFlow on constrained long-form text generation using the Qwen2.5 (Team, 2024) series models fine-tuned under our workflow-optimized framework. Comparisons are made with GPT-4o-mini (OpenAI, 2024) and with the CogWriter method applied to both Qwen2.5 (Team, 2024) and GPT-4o-mini (OpenAI, 2024). Evaluation focuses on constraint satisfaction, coherence, and overall quality. Results demonstrate that GenFlow consistently outperforms all baselines, validating the effectiveness of integrating constraint signals into workflow optimization.

## 2 RELATED WORK

**Constrained Long-Form Text Generation.** Constrained Long-Form Text Generation. Existing efforts seek to scale LLMs toward coherent ultra-long outputs under constraints. LongWriter (Bai & et al., 2024) extends generation beyond 10k words by combining an agent-based decomposition pipeline (AgentWrite) with the LongWriter-6k dataset and DPO (Rafailov et al., 2024), though it depends heavily on large curated data and still lacks workflow-level optimization. In parallel, Cog-Writer (Wan & et al., 2025) introduces a cognitive-inspired multi-agent framework that effectively mimics human writing strategies such as planning, reviewing, and monitoring (Flower & Hayes, 1981; Hayes & Flower, 2016; Kellogg, 2013; Bereiter & Scardamalia, 2013). While effective for maintaining coherence, its iterative reviews incur high cost and the absence of explicit reward signals limits adaptability (Yao et al., 2023a; Wu et al., 2024b; Zhang et al., 2024). Together, these

studies highlight both the potential and challenges of constrained long-form generation, motivating approaches that tightly couple structured planning with dynamic feedback.

**Agentic Workflow.** Beyond long-form generation, agentic workflows have been explored to enhance planning, coordination, and adaptivity. For process automation, multi-agent workflows with execution memory and fault tolerance are discussed in (Li et al., 2021), while non-autoregressive workflows for tool-augmented interactions are explored in (Gu et al., 2021). Dynamic hierarchical workflows optimized through self-reflection are introduced in (Wang et al., 2022), and systems like AutoPlanner (Zhao et al., 2021) focus on task planning optimization. Reinforcement learning for workflow adaptation is studied in (Liang et al., 2020), and constraint-based optimization techniques are explored in (Zhang et al., 2022b). These approaches highlight the flexibility of agentic workflows in diverse applications, but overlook challenges specific to constrained long-form text generation.

## 3 PRELIMINARIES

In this section, we introduce key foundations of the GenFlow, including constrained long-form text generation, rollout-based reward evaluation, and Direct Preference Optimization (DPO).

**(i) Constrained Long-Form Text Generation.** The goal is to generate coherent, fluent, and contextually relevant text that fully satisfies task-specific requirements, including constraints on content, structure, and writing style. Formally, given a task prompt $x$ and constraints $T$, the output sequence $Y = (y_1, \ldots, y_n)$ should satisfy (Liu et al., 2023; Wu et al., 2022; Zhang et al., 2023). Techniques to control and constrain text generation have been explored in recent works (Tan et al., 2023; Yu et al., 2022; Li et al., 2023), including approaches that focus on fine-tuning models for task-specific constraints (Wu et al., 2022) and incorporating structural constraints (Li et al., 2023).

$$\mathcal{L}(Y \mid x, T) \leq \mathcal{L}(Y' \mid x, T), \quad \forall Y' \neq Y, \tag{1}$$

where $x$ denotes the task prompt; $T = \{T_{\text{single}}, T_{\text{range}}, T_{\text{periodic}}\}$ is the set of task constraints, with $T_{\text{single}}$, $T_{\text{range}}$, and $T_{\text{periodic}}$ denoting single-form, range-based, and periodic constraints respectively. $Y = (y_1, \ldots, y_n)$ is the generated output sequence. $Y'$ is any alternative sequence, and $\mathcal{L}(\cdot \mid x, T)$ is the loss function measuring constraint satisfaction.

**(ii) Rollout-Based Reward Evaluation.** To systematically evaluate candidate text segments or sub-plans, we employ a rollout-based simulation approach, where the reward of a partial output $z$ is estimated as (He et al., 2022; Brown et al., 2020; Chen et al., 2023). Rollout-based methods have been widely used in reinforcement learning for evaluating sequence generation (Silver et al., 2021; Smith et al., 2020), where they estimate the potential long-term reward by simulating future steps. These techniques have also been adapted for long-form text generation tasks to improve evaluation efficiency and generate more accurate rewards (Li et al., 2022; Zhang et al., 2021).

$$r(z) = \frac{1}{N} \sum_{i=1}^{N} R\big(z \oplus \hat{Y}_{\text{rollout}}^{(i)}\big), \tag{2}$$

where $N$ denotes the number of rollouts, $R(\cdot)$ is an evaluation function measuring aspects like constraint satisfaction and coherence, and $z \oplus \hat{Y}_{\text{rollout}}^{(i)}$ denotes concatenation of $z$ with its $i$-th rollout.

**(iii) Direct Preference Optimization (DPO).** Preference alignment via DPO (Rafailov et al., 2024) compares candidate outputs $z^+$ and $z^-$ with a sigmoid function over their reward difference:

$$P(z^+ > z^- \mid x) = \sigma(r(z^+) - r(z^-)). \tag{3}$$

The training objective encourages $\pi_\theta$ to favor higher-reward outputs while regularizing against $\pi_{\text{ref}}$:

$$\mathcal{L}_{\text{DPO}}(\theta) = -\mathbb{E}_{(z^+, z^-) \sim D}\Big[ \log \sigma\big(\beta \log \frac{\pi_\theta(z^+|x)}{\pi_{\text{ref}}(z^+|x)} - \beta \log \frac{\pi_\theta(z^-|x)}{\pi_{\text{ref}}(z^-|x)}\big)\Big]. \tag{4}$$

where $z^+$ and $z^-$ are higher- and lower-preference outputs, $r(\cdot)$ is the reward, $\sigma$ the sigmoid, $\pi_\theta$ the policy, $\pi_{\text{ref}}$ the reference model, $\beta$ a temperature, and $(z^+, z^-) \sim D$ pairs from the dataset.

## 4 METHODOLOGY

In this section, as shown in Figure 3, we introduce GenFlow, which includes Constraint-Aware Planning, Binary Relevance Filtering, and Reward-Guided Optimization.

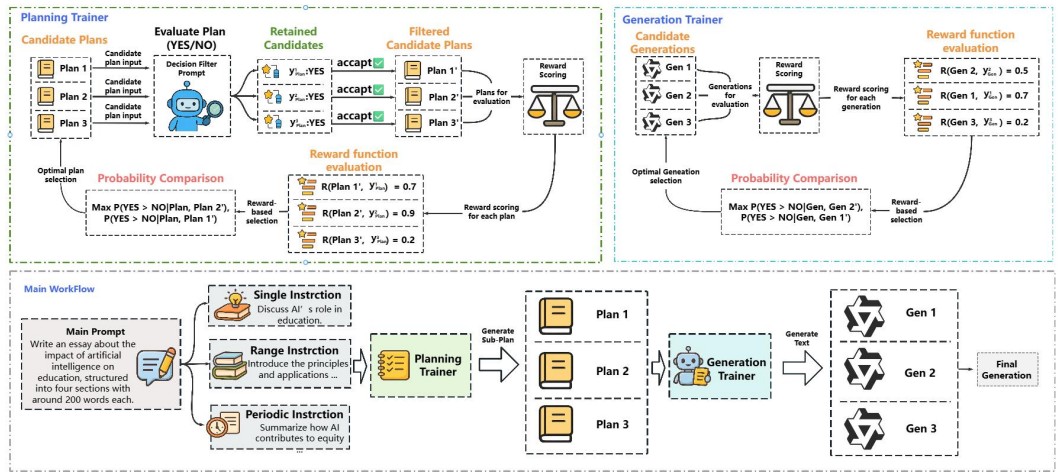

Figure 3: Overview of the GenFlow framework: a constrained generation process guided by reward signals, ensuring high-quality outputs, task-specific coherence, and structural consistency.

## 4.1 CONSTRAINT-AWARE GLOBAL PLANNING

As shown in the Planning Trainer of Figure 3, this stage trains the planning module to generate high-quality hierarchical plans via candidate generation, constraint-aware revision, and top-K selection.

The Global Planning module generates $N$ candidate hierarchical plans from a task-specific prompt $p_{\text{plan}}$, global constraints $T$, and context $C$. Each plan is represented as an ordered sequence $p_i = \{s_{i,1}, \ldots, s_{i,m_i}\}$, where each sub-plan $s_{i,j}$ specifies a localized planning unit. For generation, we define $T_{i,j}^{\text{local}} \subseteq T$ as the constraints relevant to $s_{i,j}$, and $C_{i,j}^{\text{local}} \subseteq C$ as the context available at step $j$, including information derived from the prefix $s_{i,1:j-1}$. Under this formulation, the model generates each plan hierarchically and autoregressively:

$$\Pr(p_i \mid p_{\text{plan}}, T, C; \theta) = \prod_{j=1}^{m_i} \Pr\left(s_{i,j} \mid s_{i,1:j-1}, p_{\text{plan}}, T_{i,j}^{\text{local}}, C_{i,j}^{\text{local}}; \theta\right), \quad i = 1, \ldots, N. \quad (5)$$

This factorization follows from assuming that each sub-plan is conditionally independent of non-prefix elements given its preceding sub-plans and their induced local constraints and context. The parameter set $\theta$ denotes the model parameters. Where $s_{i,j}$ is the $j$-th sub-plan of $p_i$, $T_{i,j}^{\text{local}} \subseteq T$ and $C_{i,j}^{\text{local}} \subseteq C$ are local constraints and context for $s_{i,j}$, $m_i = |p_i|$ is the number of sub-plans, and $\theta$ are model parameters for generation. A binary indicator ensures each candidate plan adheres to constraints and maintains valid hierarchical structure, identifying plans fully complying with logical organization and task requirements:

$$\mathbf{1}_{\text{valid}}(p_i) = \begin{cases} 1, & \text{if } p_i \text{ satisfies all constraints and hierarchical structure,} \\ 0, & \text{otherwise.} \end{cases} \quad (6)$$

where $\mathbf{1}_{\text{valid}}(p_i)$ equals 1 if the plan satisfies all constraints and hierarchical structure, 0 otherwise. Plans violating constraints or structure undergo a probabilistic revision process, where each plan is updated to increase the likelihood of satisfying constraints while preserving coherence:

$$\Pr(p_i^{\text{revise}} \mid p_i, p_{\text{plan}}, T, C; \theta) \propto \mathbf{1}(p_i^{\text{revise}}) \cdot \Pr(p_i^{\text{revise}} \mid p_i, p_{\text{plan}}; \theta), \quad (7)$$

where $p_i^{\text{revise}}$ is the revised plan, $\mathbf{1}(p_i^{\text{revise}})$ as the binary indicator, and $p_i$ is the original plan. A refinement step produces the final set of candidate plans, $P_{\text{final}} = \{p_i^{\text{final}}\}$, ensuring structure. Top-$K$ selection (Luo et al., 2025b) identifies highest-quality plans for generation.

$$P_{\text{top}} = \text{TopK}\left(\{f_{\text{score}}(p_i^{\text{final}}) \mid p_i^{\text{final}} \in P_{\text{final}}\}, K\right), \quad (8)$$

where $P_{\text{top}}$ is the set of selected plans and $f_{\text{score}}(\cdot)$ evaluates constraint satisfaction and coherence.

**Proposition 1.** *Constraint-aware hierarchical global planning improves structural validity and ensures constraint satisfaction in candidate plans.*

*Proof.* We provide experimental results in Section 5.4 and theoretical proofs in Appendix B.1. □

## 4.2 BINARY RELEVANCE FILTERING

As shown in the Planning Trainer of Figure 3, this evaluator filters plans by checking task constraints and structural requirements, retaining only those passing the binary evaluation.

After iterative refinement and top-$K$ selection, the global plans $P_{\text{final}} = \{p_1, \ldots, p_N\}$ undergo a binary relevance check to ensure they satisfy task requirements and maintain semantic quality beyond structural and constraint validation. Let $p_{\text{filter}}$ denote the template prompt defining six evaluation criteria—relevance, completeness, coherence, efficiency, specificity, and consistency (Appendix A.3). We then assign each candidate plan a binary indicator:

$$\delta_i = \mathbf{1}_{\text{relevant}}(p_i) = \begin{cases} 1, & \text{if } p_i \text{ satisfies all six criteria in } p_{\text{filter}}, \\ 0, & \text{otherwise}, \end{cases} \quad i = 1, \ldots, N, \quad (9)$$

where $\delta_i$ denotes whether the $i$-th candidate passes the relevance check, $p_i \in P_{\text{final}}$ is the refined plan from the Global Planning module, and $p_{\text{filter}}$ specifies the template for semantic and contextual evaluation. Plans with $\delta_i = 1$ proceed to sub-plan decomposition and generation, while those with $\delta_i = 0$ are discarded or optionally regenerated. The relevance check acts as a Bernoulli variable indicating whether each plan meets all required criteria, strengthening the filtering process.

$$\delta_i \sim \text{Bernoulli}\Big( \Pr(\delta_i = 1 \mid p_i, p_{\text{filter}}) \Big), \quad (10)$$

where $\delta_i$ is the binary indicator for the $i$-th plan, $p_i \in P_{\text{final}}$ is the refined plan, and $p_{\text{filter}}$ encodes six criteria. The probability $\Pr(\delta_i = 1 \mid p_i, p_{\text{filter}})$ is computed as follows:

$$\Pr(\delta_i = 1 \mid p_i, p_{\text{filter}}) = \prod_{k=1}^{6} p_{i,k}, \quad (11)$$

where $p_{i,k}$ denotes the probability that plan $p_i$ satisfies the $k$-th criterion under $p_{\text{filter}}$, and the product over $k = 1, \ldots, 6$ aggregates these independent criterion-wise probabilities. This formulation interprets $\delta_i$ as the joint likelihood that a plan meets all required constraints. The filtered set of plans is then obtained as

$$P_{\text{filtered}} = \{p_i \in P_{\text{final}} \mid \delta_i = 1\}, \quad (12)$$

where $P_{\text{filtered}} = \{p_i \in P_{\text{final}} \mid \delta_i = 1\}$ denotes the set of candidate plans that passed the relevance check, ensuring that only plans satisfying all criteria are retained for reward-guided sub-plan optimization, while plans with $\delta_i = 0$ are discarded or regenerated for further review.

**Proposition 2.** *Template-driven binary relevance filtering significantly improves overall plan quality by discarding suboptimal candidates.*

*Proof.* We provide experimental results in Section 5.4 and theoretical proofs in Appendix B.2. □

## 4.3 REWARD-GUIDED OPTIMIZATION

After binary relevance filtering, each plan is decomposed into sub-plans and subsequently expanded into text segments. Both stages are trained within a unified reward-guided framework. For any candidate $X$ (a sub-plan $S_j$ or a segment $G_k$), we compute a rollout-based reward:

$$r(X) = \frac{1}{N} \sum_{i=1}^{N} R\big(X \oplus \widehat{X}_{\text{rollout}}^{(i)}, \ p_{\text{reward}}\big), \quad (13)$$

where $\widehat{X}_{\text{rollout}}^{(i)}$ is the $i$-th rollout continuation generated by the *same model parameters* $\theta$, $R(\cdot, p_{\text{reward}})$ is the reward function conditioned on the evaluation prompt, and $\oplus$ denotes concatenation. Since rollouts depend on $\theta$, the reward itself becomes a parameter-dependent signal, enabling end-to-end optimization.

Given two candidates $X^+$ and $X^-$ with different rewards, their preference is defined as:

$$P(X^+ \succ X^-) = \sigma\big(r(X^+) - r(X^-)\big), \quad (14)$$

where $\sigma$ is the sigmoid function. This preference feeds directly into the DPO objective:

$$\mathcal{L}_{\text{DPO}}(\theta) = -\mathbb{E}_{(X^+, X^-) \sim D} \left[ \log \sigma \bigg( \beta \log \frac{\pi_\theta(X^+ \mid x)}{\pi_{\text{ref}}(X^+ \mid x)} - \beta \log \frac{\pi_\theta(X^- \mid x)}{\pi_{\text{ref}}(X^- \mid x)} \bigg) \right], \quad (15)$$

Table 1: Accuracy comparison of Qwen and GPT variants and GenFlow models.

| Model / Method | Accuracy Once | Accuracy Range | Accuracy Periodic | Average Accuracy |
|---|---|---|---|---|
| qwen2.5-0.5B-Instruct | 0.2346 | 0.2632 | 0.1176 | 0.2051 |
| + Cogwriter | 0.6754 | 0.6294 | 0.6029 | 0.6359 |
| **+ GenFlow** | **0.5847** | **0.725** | **0.6105** | **0.6401** |
| qwen2.5-1.5B-Instruct | 0.3445 | 0.3761 | 0.1136 | 0.2781 |
| + Cogwriter | 0.4739 | 0.5768 | 0.3741 | 0.4749 |
| **+ GenFlow** | **0.5315** | **0.4899** | **0.4059** | **0.4758** |
| qwen2.5-7B-Instruct | 0.4379 | 0.4807 | 0.283 | 0.4005 |
| + Cogwriter | 0.5108 | 0.5163 | 0.3499 | 0.459 |
| **+ GenFlow** | **0.7059** | **0.6173** | **0.6443** | **0.6558** |
| gpt-4o-mini | | | | |
| + Cogwriter | 0.4918 | 0.4026 | 0.2386 | 0.3777 |

Table 2: Ablation study of the trained Planning and Generation modules in GenFlow.

| Method | Accuracy Once | Accuracy Range | Accuracy Periodic | Average Accuracy |
|---|---|---|---|---|
| Base Model (Qwen2.5-7B) | 0.6316 | 0.3714 | 0.2 | 0.401 |
| + Planning only | 0.5714 | 0.317 | 0.2045 | 0.3643 |
| + Generation only | 0.5172 | 0.5455 | 0.35 | 0.5142 |
| **Full GenFlow (Planning + Generation)** | **0.8485** | **0.6667** | **0.44** | **0.6517** |

where $D$ is the pairwise preference dataset, $\pi_\theta$ the learned policy, $\pi_{\text{ref}}$ the reference model, and $\beta$ controls preference sharpness. This contrastive optimization aligns sub-plan selection and segment generation with reward objectives, improving structure, coherence, and constraint satisfaction.

**Proposition 3.** *Reward-guided optimization effectively aligns sub-plans and segments with task-specific objectives and quality through preference modeling.*

*Proof.* See empirical evidence in Section 5.5 and theoretical analysis in Appendix B.3. □

## 4.4 SEGMENT-LEVEL GENERATION TRAINER

As shown in the Generation Trainer of Figure 3, this stage trains the generation module to expand hierarchical plans into coherent, constraint-aligned text using autoregressive reward-guided learning.

Given a final plan $p = \{u_1, \ldots, u_L\} \in P_{\text{top}}$, the model generates the textual segment $g_k$ for each sub-plan $u_k$ following an autoregressive factorization:

$$\Pr(g_k \mid u_k, C_k;\ \phi) = \prod_{t=1}^{|g_k|} \Pr\left(g_k^{(t)} \mid g_k^{(<t)},\ u_k,\ C_k;\ \phi\right), \tag{16}$$

where $g_k$ is the generated segment, $g_k^{(t)}$ the $t$-th token, $g_k^{(<t)}$ the prefix, $C_k \subseteq C$ the local context, and $\phi$ the generator parameters; a rollout-based reward evaluates each segment's quality and utility:

$$\mathcal{R}(g_k) = \frac{1}{M} \sum_{m=1}^{M} \mathcal{J}\left(g_k \oplus \widehat{G}^{(m)},\ \mathcal{Q}_{\text{seg}}\right), \tag{17}$$

where $\widehat{G}^{(m)}$ is the $m$-th continuation sampled from the model, $\mathcal{J}(\cdot)$ is the evaluation function, $\mathcal{Q}_{\text{seg}}$ is the segment-level scoring prompt, and $\oplus$ denotes concatenation. Given two segment candidates $g^+$ and $g^-$ for the same sub-plan, their preference probability is computed as:

$$P(g^+ \succ g^-) = \sigma\left(\mathcal{R}(g^+) - \mathcal{R}(g^-)\right), \tag{18}$$

where $\sigma(\cdot)$ is the sigmoid function. The Generation Trainer is optimized via a DPO-style objective:

$$\mathcal{L}_{\text{gen}}(\phi) = -\mathbb{E}_{(g^+, g^-) \sim \mathcal{D}_{\text{seg}}} \left[ \log \sigma\left( \beta \log \frac{\pi_\phi(g^+ \mid u_k)}{\pi_{\text{ref}}(g^+ \mid u_k)} - \beta \log \frac{\pi_\phi(g^- \mid u_k)}{\pi_{\text{ref}}(g^- \mid u_k)} \right) \right], \tag{19}$$

where $\mathcal{D}_{\text{seg}}$ is the set of segment-level preference pairs, $\pi_\phi$ is the learned generator, $\pi_{\text{ref}}$ is the reference model, and $\beta$ is a temperature parameter controlling preference sharpness.

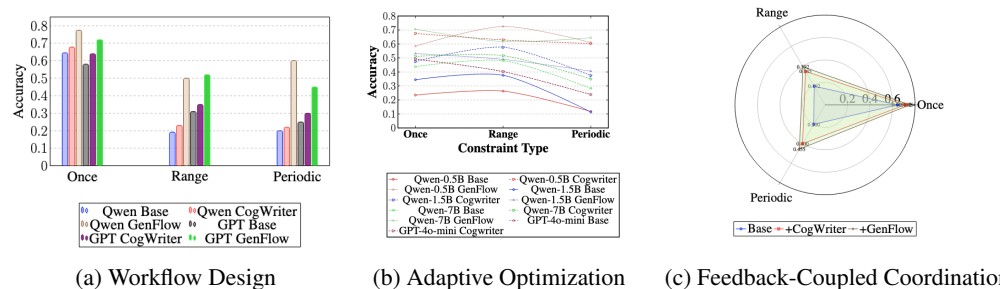

(a) Workflow Design      (b) Adaptive Optimization      (c) Feedback-Coupled Coordination

Figure 4: Comparison of workflow enhancements including constraint-aware design (RQ3), adaptive optimization (RQ4), feedback-coupled coordination (RQ5).

## 4.5 MAIN WORKFLOW

Figure 3 summarizes GenFlow's end-to-end workflow. During training, global plans are produced, filtered, optimized with reward-guided updates, and expanded into textual segments via the generation module. In inference, the system generates and filters plans, producing the final output by executing the chosen plan, ensuring structural coherence and constraint satisfaction throughout.

## 5 EXPERIMENTS

This section presents the experimental setup, main results, and analysis of our approach. We address the following research questions (RQs): **RQ1:** How much does GenFlow improve model performance over baseline approaches? **RQ2:** How effective are GenFlow's key components, as shown by ablation studies? **RQ3:** How does constraint-aware workflow design enhance constraint satisfaction? **RQ4:** How does adaptive workflow optimization improve robustness under evolving constraints? **RQ5:** How does feedback-coupled coordination improve generation quality?

## 5.1 EXPERIMENTAL SETUP

**Datasets.** Training and test sets follow **LongGenBench-16K** (Wu et al., 2024a; Liu et al., 2022; Wang et al., 2023; Zhang et al., 2021) for long-form generation tasks challenging reasoning (Shao et al., 2019; Liu et al., 2023). Four scenarios assess temporal consistency (**Diary, Menu**) and spatial reasoning (**Skyscraper, Urban**) (Xu et al., 2021; Liu et al., 2021; He et al., 2021; Zhang et al., 2023; Wang et al., 2020). Details in Appendix D.

**Baseline.** We compare GenFlow with CogWriter (Wan & et al., 2025; Wang et al., 2023) and GPT-4o-mini (OpenAI, 2024) across three Qwen2.5 (Team, 2024) scales: 0.5B, 1.5B, 7B. Details in Appendix E.

**Evaluation Metrics.** We evaluate model performance from LongGenBench with four metrics. Instruction Following Accuracy measures adherence to single (**Acc. Once**) (Li et al., 2022), range (**Acc. Range**) (Zhou et al., 2021), and periodic (**Acc. Periodic**) instructions (Tan et al., 2022), with their average reported as **Avg. Acc**, providing a measure of task completion (Huang et al., 2021; Zhang et al., 2020). More details are in Appendix F.

**Implementation Details.** We implement GenFlow and all baselines following the released **LongGenBench** (Wu et al., 2024a; Liu et al., 2022) framework, incorporating its pre-configured components for efficient model training and optimization (Chen et al., 2020; Liu et al., 2021). For training and evaluation, all experiments are conducted on 3 NVIDIA A40 GPUs (48GB), enabling high-throughput parallel computation and fast processing times across multiple experiments (Xu et al., 2021; Zhang et al., 2023). More details are provided in Appendix G.

## 5.2 MAIN RESULTS (RQ1)

As shown in Table 1, we compare GenFlow with baselines across different base models, and observe that GenFlow consistently outperforms all baselines. We highlight two key findings.

**Workflow-Driven Optimization Improves Constraint Satisfaction and Generation Quality.** Direct generation and CogWriter often fail to satisfy complex constraints, leading to suboptimal accuracy. GenFlow integrates planning, filtering, and reward-guided training into a unified workflow, significantly boosting accuracy. For example, under Qwen2.5-7B-Instruct, the base model achieves 0.3458, CogWriter yields 0.3444, while GenFlow raises it to 0.4774. This demonstrates that GenFlow's structured workflow effectively enforces constraints while maintaining coherence.

**Larger Base Models Amplify GenFlow Benefits.** As base model size increases from 0.5B to 7B, GenFlow achieves higher accuracy across all constraint types. With Qwen2.5-7B-Instruct, GenFlow attains 0.7692 on once instructions and 0.4231 on range instructions, outperforming base model and CogWriter. Larger models better exploit GenFlow's structural signals, enhancing constraint adherence and generation quality.

### 5.3 Ablation Study on the Effectiveness of GenFlow's Key Components (RQ2)

As shown in Table 2, we conduct ablation experiments on the trained Planning and Generation modules of GenFlow using Qwen2.5-7B-Instruct as the base model. We have three main observations.

**Larger Base Models Amplify GenFlow Benefits.** As base model size increases from 0.5B to 7B, GenFlow achieves higher accuracy across all constraint types. With Qwen2.5-7B-Instruct, GenFlow attains 0.7692 on single-step instructions and 0.4231 on range instructions, outperforming both the base model and CogWriter. Larger models can more effectively exploit GenFlow's structural signals, enhancing constraint adherence and generation quality.

**Training the Generation Module Yields Stronger Improvements.** When only the Generation module is trained, the average accuracy further increases to 0.4192. This suggests that reward-guided contrastive optimization at the segment level is crucial for ensuring textual coherence and constraint satisfaction, thereby directly enhancing the final outputs. The gains are more pronounced in both once and periodic instructions, particularly under complex constraints.

**Joint Training Achieves the Best Performance.** When both modules are trained together, GenFlow achieves the highest average accuracy of 0.4774, significantly surpassing the base model (0.3458). This confirms that the two modules are complementary: the Planning module improves structural organization and constraint coverage, while the Generation module enforces segment-level quality and coherence. The combination of both enables GenFlow to consistently produce high-quality, constraint-compliant long-form text across all instruction types.

We note that the "Planning-only" variant disables the constraint-aware DPO refinement in the Generation module. In this setting, although the model receives a hierarchical plan, it is not optimized to follow it, often leading to plan–generation mismatch and thus lower accuracy than the base model. This behavior is expected and indicates that planning alone is insufficient; strong constraint adherence emerges only when planning is paired with generation-level refinement. The superior performance of the full GenFlow model therefore reflects the complementary roles of the two modules.

### 5.4 Constraint-Aware Workflow Design (RQ3)

As shown in Figures 5 and 4a, constraint-aware workflow design enhances model performance across different constraint types (Once, Range, Periodic). The comparison between models with and without GenFlow shows improvements, particularly under challenging constraints like *Range* and *Periodic*. These results demonstrate that GenFlow prevents performance degradation often seen with naive approaches, especially with complex constraints.

**Workflow Integration Improves Accuracy.** GenFlow's positive impact is not limited to a single instruction type or constraint (Luo et al., 2025c). Across all three constraint types, we observe con-

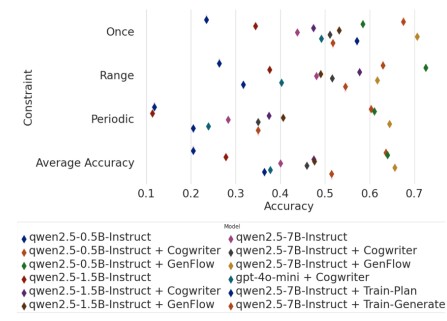

Figure 5: Constraint Accuracy Comparison.

sistent performance improvements, especially in more difficult constraint settings like *Range* and *Periodic*. The performance boost highlights how GenFlow's workflow integration strengthens both local (constraint-specific) compliance and global (overall model accuracy) improvement.

**Generalizable Across Constraint Types.** These findings confirm that GenFlow addresses the core challenge (iii) outlined, showing that constraint-aware workflow integration is a robust and generalizable solution. By improving compliance with the defined constraints, GenFlow enhances model accuracy across various use cases, particularly in tasks that require high accuracy and efficiency, such as long-form generation.

## 5.5 ADAPTIVE WORKFLOW OPTIMIZATION (RQ4)

As shown in Figures 4b and the subsequent performance charts, adaptive workflow optimization plays a crucial role in addressing the evolving complexities of constraint types, such as "Once," "Range," and "Periodic." The experiments clearly validate the need for dynamic optimization in response to evolving task requirements.

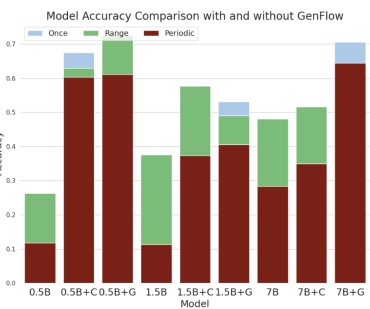

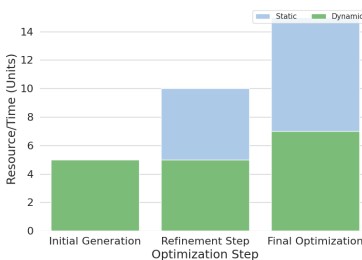

(a) Constraint-Aware Workflow Design    (b) Adaptive Workflow Optimization

Figure 6: Comparison of workflow design and optimization strategies.

**GenFlow Sustains Performance Under Complexity.** As shown in Figure 6a, while baseline models and static methods (e.g., without GenFlow) show a sharp decline in accuracy as constraints increase in complexity, GenFlow maintains stable performance across all constraint types. This demonstrates that GenFlow's adaptive optimization mitigates the brittleness of static approaches, ensuring reliable results under challenging conditions.

**Refinement Cost Comparison: Static vs. Dynamic Filtering.** The Figure 6b highlights the efficiency of dynamic optimization through GenFlow. Static methods, which rely on fixed datasets and costly refinement steps, incur significantly higher costs as the complexity increases. In contrast, GenFlow's dynamic approach, shown in the "Dynamic Filtering" bar, ensures that the system adapts with minimal increase in computational or resource cost, making it a much more resource-efficient solution. This validates that GenFlow's ability to adaptively optimize based on the task's evolving needs can significantly reduce resource consumption compared to traditional static methods.

**Refinement Cost Comparison: Static vs. Dynamic Filtering.** The "Refinement Cost Comparison" chart highlights the efficiency of dynamic optimization through GenFlow. Static methods, relying on fixed datasets and costly refinement steps, incur significantly higher costs as complexity increases. In contrast, GenFlow's dynamic approach, shown in the "Dynamic Filtering" bar, ensures the system adapts with minimal increase in computational or resource cost, making it a more resource-efficient solution. This validates that GenFlow's ability to adaptively optimize based on the task's evolving needs can reduce resource consumption compared to traditional static methods.

## 5.6 FEEDBACK-COUPLED COORDINATION (RQ5)

The line plot in Figure 4c illustrates how the performance of different models evolves across the four training stages—initialization, planning, generation, and refinement—shedding light on the impact of feedback-coupled coordination.

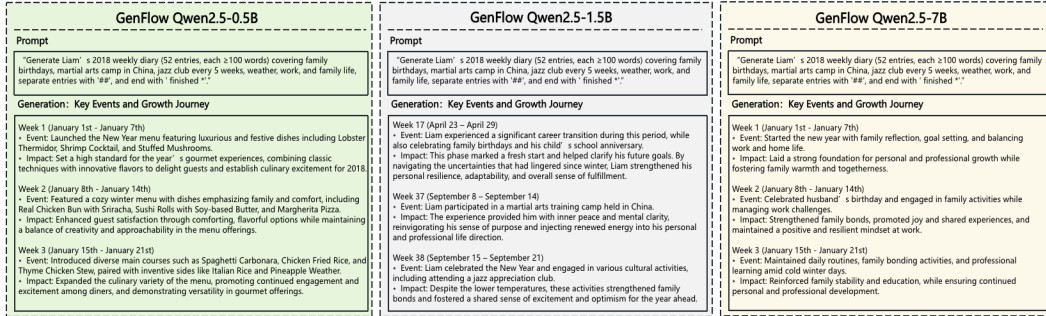

Figure 7: Case study on generation quality under a representative constraint prompt, comparing Gen-Flow with Qwen2.5-0.5B-Instruct, Qwen2.5-1.5B-Instruct, and Qwen2.5-7B-Instruct base models.

**Feedback Coupling Leads to Progressive Gains.** The Full GenFlow model shows steady improvement across all four stages, benefiting from real-time feedback coupling. This feedback ensures continuous refinement and avoids fluctuations seen in models without coupling, which rely on isolated planning or generation stages (Luo et al., 2025a). Feedback coupling actively enhances performance over time.

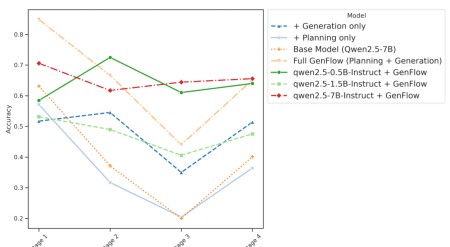

**Coordinated Improvement in Training Stages.** Full GenFlow maintains balanced performance across stages, with significant improvements in Stage 4. In contrast, models without feedback coupling suf-

Figure 8: Model Accuracy Across Different Training Stages.

fer performance drops, particularly between Stage 2 and 3, indicating poor coordination. Full GenFlow effectively integrates feedback, ensuring smooth transitions and continuous improvement. Traditional models struggle to adapt to evolving constraints. As shown in Figure 8, Full GenFlow addresses this by incorporating feedback at each stage, preventing degradation as constraints grow more complex.

## 5.7 CASE STUDY: PROMPT-BASED GENERATION ACROSS QWEN2.5 VARIANTS (RQ6)

We present a comparative case study using three Qwen2.5 variants (0.5B, 1.5B, 7B) under identical prompts to generate coherent life narratives. As shown in Figure 7, the 7B model exhibits superior narrative depth and consistency compared to smaller variants, demonstrating enhanced reasoning and contextual awareness. This highlights the importance of model scaling in complex generative tasks requiring sustained coherence.

## 6 CONCLUSION

In this work, we introduce GenFlow, a workflow-optimized framework for constrained long-form text generation. By decomposing the writing objective into structured sub-plans, applying binary filtering and reward-guided optimization, and generating text segment by segment, GenFlow tightly integrates planning and generation under a unified reinforcement and preference learning paradigm. A consistent rollout-based reward mechanism ensures that only high-quality sub-plans and generations are reinforced, while low-quality outputs are suppressed. Experiments across diverse long-context benchmarks demonstrate that GenFlow outperforms strong baselines in constraint satisfaction, coherence, and overall generation quality.

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

APPENDIX

# A    PROMPTS USED IN GENFLOW

## A.1    PLAN CONSTRUCTION PROMPT

**Week Plan Prompt.** The Week Plan Construction Prompt guides the model to create a 52-week plan based on user requirements. It identifies special events and their exact weeks, including periodic events. The model generates a structured weekly plan in JSON format, and a revision step ensures all events strictly follow user specifications.

Figure 9: Plan Construction Prompt illustration

**Floor Plan Prompt.** The Floor Plan Prompt instructs the model to create a detailed floor-by-floor plan for a skyscraper. Special facilities are assigned to specific floors, including recurring ones, and the plan is output in JSON format.

Figure 10: Floor Plan Prompt illustration

**Menu Plan Prompt.** The Menu Plan Prompt generates a 52-week menu schedule, assigning special dishes to specific weeks, including periodic ones.

```
---- Role ----

You are an expert chef, and your task is to create a weekly menu plan containing 52 weeks.
User requirements: {prompt}
---- Goal ----

Think step by step. Analyse the user requirements to identify special dishes and their exact week and list them in
"special_dishes". If the dish is periodic, list them seperately by number and specify which weeks they will be in (e.g. 1st
[dish_name]). Then follow your analysis to create a weekly menu plan for each week.
---- Output Format ----                              ---- Return ----

Here is an example of the output format:            Return your analysis and plan in ONLY this exact json format:
{{                                                   {{
    "analysis": "",                                      "analysis": "",
    "special_dishes": [                                  "special_dishes": [
        {{                                                   {{
            "dish_name": "Winter Solstice Feast featuring Venison        "dish_name": "dish_name",
Stew",                                                       "week_numb": "date and week number",
            "week_numb": "05-13, Week 19"                        }}
        }}                                               ],
        ...                                              # Strictly follow the special_dishes for specific weeks
    ],                                                   "weekly_plan": [
    "weekly_plan": [                                         {{
        {{                                                       "week_id": "Menu Week 1 (January 1st - January 7th)",
            "week_id": "Menu Week 1 (January 1st - January 7th)",        "dishes": "Australia Day BBQ featuring Lamb Chops"
            "dishes": "Australia Day BBQ featuring Lamb Chops"        }},
        }},                                                      ...
        ...                                              ]
    ]
}}
```

Figure 11: Menu Plan illustration

**Block Plan Prompt.** The Block Plan Prompt guides the design of a 10×10 city block grid, assigning special uses to specific blocks and handling periodic patterns.

```
---- Role ----

You are an expert urban planner. Create a precise city block plan that EXACTLY matches ALL user requirements.

---- Prompt ----

 User requirements: {prompt}

---- Goal ----

CRITICAL ANALYSIS INSTRUCTIONS:
1. Identify EVERY specific building, facility, or land use mentioned
2. Note EXACT locations or block specifications
3. Pay attention to zoning requirements, special facilities, or community needs
4. Identify any special features for specific blocks
---- Return format ----

Return your analysis and plan in this EXACT format:
{{
    "analysis": "Detailed analysis of every block planning requirement in the user prompt",
    "special_blocks": [
        {{
            "block_number": "Block identifier as specified",
            "use": "Exact use/purpose from requirements",
            "special_features": "Any special features mentioned"
        }}
    ],
    "block_plan": [
        {{
            "block_id": "Block 1",
            "use": "Exact use matching user requirements"
        }}
    ]
}}
```

Figure 12: Block Plan Prompt illustration

## A.2 WRITING PROMPT

**Weekly Diary Prompt.** The Weekly Diary Prompt guides the model to write a 200-word diary entry for a given week. It incorporates weekly events, ensures coherence with the overall yearly plan, and checks user requirements for special events. The model outputs a structured JSON containing the week identifier, a reasoning check, and the diary entry text.

```
---- Role ----
 You are an expert writer.
---- Goal ----
Write a 200-word weekly diary entry for the week of {week_id}.
The events for this week are: {events}
You should consider the coherence of the diary entry referring to the plan of the whole year: {weekly_plan}
You should consider the user requirements: {prompt}
Check from the user requirements if there are any special events that should be included in the diary entry. If there are,
include them in the diary entry. If there are no special events, write a general diary entry for the week.
 ---- Return ----
Return the diary entry in the following json format:
{{
    "week_id": "{week_id}",
    "check": "reason and check if the user requirements are met",
    "diary_entry": "Your 200-word diary entry here"
}}
```

Figure 13: Weekly Diary Prompt

**Skyscraper Floor Plan Prompt.** The Skyscraper Floor Plan Prompt guides the model to design a 150-word floor plan for a specific skyscraper floor. It considers the floor's purpose, maintains coherence with the overall skyscraper plan, and verifies any special requirements from the user. The output is a structured JSON including the floor identifier, a reasoning check, and the generated floor plan.

```
---- Role ----
You are an expert designer.
---- Goal ----
Write a 150-word skyscraper floor plan for the floor of {floor_id}.
The purpose for this floor is: {purpose}
You should consider the coherence of the floor plan by referring to the plan of the whole skyscraper: {floor_plan}
You should consider the user requirements: {prompt}
Check from the user requirements if there are any special requirement that should be included in the floor plan. If
there are, include them in the floor plan. If there are no special events, write a general floor plan.
Return the floor plan for the floor of {floor_id} in the following json format:

---- Return ----
 Return the floor plan for the floor of {floor_id} in the following
 json format:
{{
    "floor_id": "{floor_id}",
    "check": "reason and check if the user requirements are met",
    "plan": "Your 150-word floor plan here"
}}
```

Figure 14: Skyscraper Floor Plan Prompt

**Weekly Menu Plan Prompt.** The Weekly Menu Plan Prompt guides the model to create a 200-word menu plan for a given week. It incorporates provided dishes, ensures coherence with the yearly menu plan, and verifies special dishes from the user requirements. The model returns a structured JSON with the week identifier, a reasoning check, and the weekly menu plan.

```
---- Role ----
   You are an expert chef.
---- Goal ----
Write a 200-word weekly menu plan for the week of {week_id}.
The dishes for this week are: {dishes_str}
You should consider the coherence of the menu plan referring to the plan of the whole year: {weekly_plan}
You should consider the user requirements: {prompt}
Check from the user requirements if there are any special dishes that should be included in the menu plan. If there are,
include them in the menu plan. If there are no special dishes, write a general menu plan for the week.

---- Return ----
Return the menu plan in the following json format:
{{
    "week_id": "{week_id}",
    "check": "reason and check if the user requirements are met",
    "week_menu": "Your 200-word menu plan here"
}}
```

Figure 15: Weekly Menu Plan Prompt

**City Block Plan Prompt.** The City Block Plan Prompt guides the model to design a 150-word city block plan. It considers the intended use of the block, ensures coherence with the overall city plan, and checks for any special user requirements. The generated output is returned in JSON format, including the block identifier, a reasoning check, and the block plan.

```
---- Role ----
You are an expert designer.
---- Goal ----
Write a 150-word city block plan for the block of {block_id}.
The use for this block is: {use}
You should consider the coherence of the block plan by referring to the plan of the whole city: {block_plan}

You should consider the user requirements: {prompt}
Check from the user requirements if there are any special requirement that should be included in the block plan. If
there are, include them in the block plan. If there are no special events, write a general block plan.

---- Return ----
Return the block plan for the block of {block_id} in the following
json format:
{{
    "block_id": "{block_id}",
    "check": "reason and check if the user requirements are met",
    "plan": "Your 150-word block plan here"
}}
```

Figure 16: City Block Plan Prompt

## A.3 BINARY DECISIONS PROMPT

As shown as Figure 17, a prompt template for binary evaluation of a plan's relevance to a given question. The judgment (YES/NO) is based on six criteria: direct relevance, completeness, logical coherence, efficiency, specificity, and consistency with the prompt context. Examples illustrate common failures—such as irrelevant steps, missing actions, or illogical order. The structured format includes the prompt, the plan, and the binary relevance label, enabling systematic assessment of planning quality.

---

**---- Goal ----**

Given the following prompt and plan, return YES if the plan is relevant to the question and NO if it isn't. A plan is considered relevant only if it meets all of the following criteria:

1. Direct Relevance: The plan must directly address the question posed in the prompt. It should not deviate from the main objective.
2. Completeness: The plan must include all necessary steps to achieve the goal described in the prompt. Missing critical steps will result in the plan being deemed irrelevant.
3. Logical Coherence: The plan must follow a logical sequence of actions. Each step should clearly lead to the next, and there should be no logical gaps or contradictions.
4. Efficiency: The plan should not include unnecessary or redundant steps. Each step must contribute directly to the final goal.
5. Specificity: The plan must be specific and actionable. Vague or overly broad steps will not be considered relevant.
6. Contextual Consistency: The plan must be consistent with the context provided in the prompt. Any steps that are inconsistent with the background information or constraints will be deemed irrelevant.

**---- Example of Irrelevant Plans ----**

- Irrelevant Content: If the prompt asks for a plan to "design an experiment to test plant photosynthesis," but the plan includes steps like "build a rocket model," this plan will be rejected.
- Incomplete Plan: If the plan omits critical steps, such as "measure the photosynthesis rate," it will be considered irrelevant.
- Illogical Sequence: If the steps in the plan do not follow a logical order, such as "measure plant growth first, then plant the seeds," the plan will be rejected.

**---- Prompt ----**

{prompts}

**---- Plan ----**

{weekly_plan} / {floor_plan} / {menu_weekly_plan} / {block_plan}

**---- Relevant ----**

(YES / NO)

Figure 17: Prompt for binary decisions

## B    THEORETICAL PROOF

### B.1    PROOF OF PROPOSITION 1

**Proposition 1.** *Constraint-aware hierarchical global planning improves structural validity and ensures constraint satisfaction in candidate plans.*

*Proof.* Let a hierarchical plan be

$$p_i = \{s_{i,1}, s_{i,2}, \ldots, s_{i,m_i}\}, \quad s_{i,j} \in \mathcal{S}, \tag{20}$$

with local constraints $T_{i,j}^{\text{local}} \subseteq T$ and local conditions $C_{i,j}^{\text{local}} \subseteq C$. Define the constraint satisfaction and structural validity indicators:

$$\mathbf{1}_T(p_i), \quad \mathbf{1}_{\text{struct}}(p_i) \in \{0, 1\}. \tag{21}$$

To make the statement precise, we introduce a mild and standard assumption on the revision operator: there exists a constant $0 < \alpha < 1$ such that for any plan $p$ produced by the generator,

$$\mathbb{E}\big[\phi(\mathcal{R}(p))\big] \le \alpha \, \mathbb{E}\big[\phi(p)\big], \tag{22}$$

where $\phi(p) \ge 0$ is a nonnegative constraint-violation score with $\phi(p) = 0$ iff $p$ satisfies all constraints and structural rules, and $\mathcal{R}$ denotes the (possibly stochastic) revision operator described in the main text. This assumption formalizes the intuition that each revision step reduces expected violation.

Let $p^{(t)}$ denote the plan after $t$ rounds of revision/refinement. By repeated application of equation 22 we obtain

$$\mathbb{E}\big[\phi(p^{(t)})\big] \le \alpha^t \, \mathbb{E}\big[\phi(p^{(0)})\big]. \tag{23}$$

Since $0 < \alpha < 1$, $\alpha^t \to 0$ as $t \to \infty$, hence

$$\lim_{t \to \infty} \mathbb{E}\big[\phi(p^{(t)})\big] = 0. \tag{24}$$

Applying Markov's inequality gives, for any $\epsilon > 0$,

$$\Pr\big(\phi(p^{(t)}) \ge \epsilon\big) \le \frac{\mathbb{E}[\phi(p^{(t)})]}{\epsilon} \xrightarrow[t \to \infty]{} 0. \tag{25}$$

Taking $\epsilon$ arbitrarily small (in particular $\epsilon = 0^+$), we conclude that

$$\Pr\big(\phi(p^{(t)}) = 0\big) \to 1 \quad \text{as } t \to \infty, \tag{26}$$

i.e. the probability that a plan is fully constraint-satisfying tends to one. Since $\phi(p) = 0$ implies both $\mathbf{1}_T(p) = 1$ and $\mathbf{1}_{\text{struct}}(p) = 1$, we obtain

$$\Pr\big[\mathbf{1}_T(p^{(t)}) = 1 \wedge \mathbf{1}_{\text{struct}}(p^{(t)}) = 1\big] \to 1, \tag{27}$$

which is the desired result: iterative constraint-aware planning yields plans that are valid with high probability. $\qquad\square$

### B.2    PROOF OF PROPOSITION 2

**Proposition 2.** *Template-driven binary relevance filtering significantly improves overall plan quality by discarding suboptimal candidates.*

*Proof.* Let $P_{\text{final}} = \{p_1, \ldots, p_N\}$ be the set of candidate plans after refinement and let

$$\delta_i = \mathbf{1}_{\text{relevant}}(p_i) \in \{0, 1\} \tag{28}$$

be the binary indicator returned by the template-driven filter (1 if the plan satisfies the six criteria, 0 otherwise). The filtered set is

$$P_{\text{filtered}} = \{p_i \in P_{\text{final}} \mid \delta_i = 1\}. \tag{29}$$

Let $Q(p)$ denote an intrinsic quality score for plan $p$ (higher is better). We make a natural and minimal assumption that the binary filter is at least weakly correlated with quality, i.e. plans that satisfy the template criteria are not on average of lower quality than those that do not:

$$\mathbb{E}\big[Q(p) \mid \delta = 1\big] \geq \mathbb{E}\big[Q(p) \mid \delta = 0\big]. \tag{30}$$

This assumption formalizes the intended behavior of the template: to retain semantically and structurally relevant, higher-quality plans.

The expected quality of the filtered set can be written as

$$\mathbb{E}\big[Q(P_{\text{filtered}})\big] = \mathbb{E}\big[Q(p) \mid \delta = 1\big]. \tag{31}$$

Similarly, the expected quality over all candidates is

$$\mathbb{E}\big[Q(P_{\text{final}})\big] = \Pr(\delta = 1)\mathbb{E}\big[Q(p) \mid \delta = 1\big] + \Pr(\delta = 0)\mathbb{E}\big[Q(p) \mid \delta = 0\big]. \tag{32}$$

Combining these expressions with equation 30 yields

$$\mathbb{E}\big[Q(P_{\text{filtered}})\big] = \mathbb{E}\big[Q(p) \mid \delta = 1\big] \geq \Pr(\delta = 1)\mathbb{E}\big[Q(p) \mid \delta = 1\big] + \Pr(\delta = 0)\mathbb{E}\big[Q(p) \mid \delta = 0\big] = \mathbb{E}\big[Q(P_{\text{final}})\big]. \tag{33}$$

Hence, the expected quality after template-driven binary relevance filtering is no smaller than the expected quality before filtering; when the filter successfully removes lower-quality plans with positive probability, the inequality is strict. This formalizes the claim that the filter improves expected plan quality by discarding suboptimal candidates. $\qquad\square$

### B.3 PROOF OF PROPOSITIONS 3

**Proposition 3.** *Reward-guided optimization effectively aligns sub-plans and segments with task-specific objectives and quality through preference modeling.*

*Proof.* After binary relevance filtering, each plan is decomposed into sub-plans for optimization, followed by content generation guided by the sub-plan. For each candidate $X$ (a sub-plan $S_j$ or a generated segment $G_k$), define its reward by rollout evaluation:

$$r(X) = \frac{1}{N} \sum_{i=1}^{N} R\big(X \oplus \widehat{X}_{\text{rollout}}^{(i)}, p_{\text{reward}}\big), \tag{34}$$

where $\widehat{X}_{\text{rollout}}^{(i)}$ are rollouts used to estimate the downstream quality of $X$. For a pair $(X^+, X^-)$ with $r(X^+) > r(X^-)$, DPO models the preference probability as

$$P(X^+ \succ X^- \mid x) = \sigma\big(r(X^+) - r(X^-)\big), \tag{35}$$

with $\sigma(z) = 1/(1 + e^{-z})$.

Recall the DPO objective used in the paper:

$$\mathcal{L}_{\text{DPO}}(\theta) = -\mathbb{E}_{(X^+, X^-) \sim D}\Big[\log \sigma\Big(\beta \log \frac{\pi_\theta(X^+ \mid x)}{\pi_{\text{ref}}(X^+ \mid x)} - \beta \log \frac{\pi_\theta(X^- \mid x)}{\pi_{\text{ref}}(X^- \mid x)}\Big)\Big]. \tag{36}$$

This objective encourages the model policy $\pi_\theta$ to increase the relative log-probability ratio for higher-reward candidates $(X^+)$ and decrease it for lower-reward candidates $(X^-)$, proportionally to the preference signal induced by the reward difference.

To see that optimizing $\mathcal{L}_{\text{DPO}}$ increases the probability of selecting higher-reward candidates, note that for a fixed pair the inner term can be viewed as a logistic regression log-likelihood where the "feature" is the log-probability ratio $\Delta_\theta(X)\beta \log \frac{\pi_\theta(X|x)}{\pi_{\text{ref}}(X|x)}$ and the label is determined by the reward difference. Gradient ascent on the expected log-sigmoid term will increase $\Delta_\theta(X^+)$ and decrease $\Delta_\theta(X^-)$ in expectation whenever $r(X^+) - r(X^-) > 0$. Concretely, the gradient of the loss w.r.t. $\Delta_\theta$ yields a term proportional to

$$\sigma\big(-\Delta_\theta(X^+) + \Delta_\theta(X^-)\big) - \sigma\big(r(X^+) - r(X^-)\big), \tag{37}$$

which is negative when $\Delta_\theta$ underestimates the reward-induced preference and thus drives updates that align $\Delta_\theta$ with the reward signal. Under regularity conditions standard in stochastic optimization

(bounded gradients, appropriate learning rates), repeated updates therefore increase the expected logit gap $\Delta_\theta(X^+) - \Delta_\theta(X^-)$, which in turn increases the model probability ratio

$$\frac{\pi_\theta(X^+ \mid x)}{\pi_\theta(X^- \mid x)} \tag{38}$$

and hence the selection probability of $X^+$ relative to $X^-$.

Under the mild Reward Smoothness assumption (see main text / Appendix) which bounds reward changes under small output perturbations, meaningful reward differences imply meaningful preference signals. Therefore, DPO optimization systematically shifts model likelihood toward higher-reward candidates, aligning generation with task-specific objectives and improving overall generation quality. □

## C  GENFLOW ALGORITHM DETAILS

The GenFlow algorithm presents an adaptive workflow optimization framework designed for generating constrained long-form content. It operates in four stages: (1) Hierarchical planning decomposes the writing prompt into sub-plans, applying constraint-aware planning for each. (2) Adaptive workflow execution generates plans, evaluates quality, and refines them based on thresholds. (3) Reward-based filtering ranks and selects the best sub-plan for the final output. (4) End-to-end policy optimization (DPO) improves the generation process by updating the policy through trajectory sampling and reward-based advantage calculation.

---

**Algorithm 1** GenFlow: Adaptive Workflow Optimization for Constrained Long-form Generation

---

**Require:** Writing prompt $p$, task-specific constraints $T$, policy $\pi_\theta$, reward function $R(\tau)$
**Ensure:** Generated long-form content $y$

1: **// 1: Hierarchical Planning**
2: Decompose writing prompt $p$ into sub-plans $P = \{p_1, p_2, \ldots, p_n\}$
3: **for** each sub-plan $p_i \in P$ **do**
4:     Apply constraint-aware planning to $p_i$
5:     Generate sub-plan: $p_i^{\text{gen}} \sim \pi_\theta(p_i \mid T)$
6:     Apply adaptive decision-making to refine $p_i^{\text{gen}}$
7: **end for**

8: **// 2: Adaptive Workflow Execution**
9: Initialize state $s_1 \leftarrow p$, trajectory $\tau \leftarrow \emptyset$
10: **for** $t = 1$ to $T$ **do**
11:     Generate adaptive plan: $\mathbf{a}_t^{\text{plan}} \sim \pi_\theta(\cdot \mid s_t)$
12:     Evaluate plan quality: $q_t = \text{Quality}(\mathbf{a}_t^{\text{plan}}, T)$
13:     **if** $q_t \geq$ threshold **then**
14:         Execute plan: $y_t = \text{Generate}(p_t, \mathbf{a}_t^{\text{plan}})$
15:         Add result to trajectory: $\tau \leftarrow \tau \cup \{(s_t, \mathbf{a}_t^{\text{plan}}, y_t)\}$
16:     **else**
17:         Refine plan: $\mathbf{a}_t^{\text{plan}} \sim \pi_\theta(\cdot \mid s_t, T)$
18:     **end if**
19: **end for**

20: **// 3: Reward-based Filtering and Decision Making**
21: Compute reward for trajectory $\tau$:

$$R(\tau) = \text{constraint satisfaction} + \text{coherence score} + \text{fidelity score}$$

22: Rank sub-plans based on reward:
$$\mathbf{a}_t^{\text{final}} = \text{Top-k}(R(\tau))$$

23: Select best plan for final output: $y \leftarrow \mathbf{a}_t^{\text{final}}$
24: **// 4: End-to-end Policy Optimization (DPO)**
25: Sample $N$ trajectories $\{\tau_i\} \sim \pi_{\theta_{\text{old}}}$
26: **for** each $\tau_i$ **do**
27:     Compute reward:
$$R(\tau_i) = R_{\text{format}}(\tau_i) + R_{\text{quality}}(\tau_i)$$

28:     Compute advantage:
$$\hat{A}(\tau_i) = \frac{R(\tau_i) - \text{mean}(\{R(\tau_j)\})}{\text{std}(\{R(\tau_j)\})}$$

29: **end for**
30: Update policy via DPO:

$$\mathcal{J}_{\text{DPO}} \sim \sum_{i=1}^{N} \sum_{t=1}^{|\tau_i|} \min\left(\rho_\theta(a_t^{(i)})\hat{A}(\tau_i), \text{clip}(\rho_\theta(a_t^{(i)}), 1 \pm \epsilon)\hat{A}(\tau_i)\right)$$

31: where

$$\rho_\theta(a_t^{(i)}) = \frac{\pi_\theta(a_t^{(i)} \mid s_{t-1}^{(i)})}{\pi_{\theta_{\text{old}}}(a_t^{(i)} \mid s_{t-1}^{(i)})}$$

---

## D  DATASET DETAILS

Our dataset is constructed based on the official LongGenBench codebase, following the methodology described in Section 3 of the main paper. It comprises **400 synthetic long-form planning and generation tasks**, spanning diverse domains such as weekly event planning, architectural floor designs, menu organization, and city block descriptions. Each task instance is automatically paired with structured prompts, hierarchical global plans, task-specific constraints (single, range, or periodic), and evaluation prompts, enabling the dataset to jointly capture global planning information and local generation requirements. Each sample includes **(i)** a global hierarchical plan, **(ii)** embedded structural constraints, **(iii)** sub-plan prompts used for preference optimization, and **(iv)** the final generation targets. On average, global plans contain 250–300 words, while individual generated segments contain 150–200 words depending on the task type.

Although the dataset is synthetically generated, this construction is both appropriate and necessary for our benchmark. Real-world human-authored corpora rarely contain explicit structural constraints—such as hierarchical outlines, length-range specifications, and periodic content formats—at the scale and diversity required for systematic evaluation. Synthetic generation enables precise and reproducible control over constraint complexity, ensuring consistent coverage across task categories without compromising the realism of the structural patterns. Importantly, the dataset serves as a *task specification* rather than a corpus for style imitation: GenFlow learns to follow constraints, not to mimic LLM-produced writing styles. Since the model never encounters test instances during training and DPO relies solely on preference signals rather than memorizing dataset patterns, the synthetic origin does not introduce leakage or closed-loop issues. Moreover, the structural constraints modeled in this dataset—such as multi-level sectioning, fixed-length segments, and periodic organization—naturally occur in real-world long-form writing, including reports, educational documents, and technical manuals, making the resulting capabilities directly transferable to human-authored scenarios.

This dataset is used for both training and evaluation under controlled settings, and all 400 samples are included when comparing different model backbones (Qwen2.5, GPT-4o-mini) as reported in Section 5. To ensure reproducibility, we adopt identical seeds and sampling parameters to those in the official LongGenBench implementation. Since all samples are generated from synthetic prompts and LLM-produced outputs, the dataset contains no personally identifiable information (PII) or sensitive content and is intended solely for academic research on long-form planning and generation.

## E  BASELINE DETAILS

**GPT-4o-mini.** In our experiments, we evaluate the proposed **GenFlow** framework against several strong baselines across different model backbones and task scenarios. For models based on `GPT-4o-mini`, we include **CogWriter** (Wan & et al., 2025), a multi-step cognitive planning and generation framework specifically designed for long-form text. CogWriter performs hierarchical planning to structure content and leverages GPT-4o-mini for each generation step, allowing it to handle complex long-form tasks with temporal and spatial constraints.

**Qwen2.5.** For models based on `Qwen2.5` across multiple scales (0.5B, 1.5B and 7B), we also compare against **CogWriter** (Wan & et al., 2025). In this setting, CogWriter utilizes Qwen2.5 as the generation backbone, employing a hierarchical planning strategy and sub-plan guided generation to produce coherent outputs for long-form scenarios such as diary writing, menu design, skyscraper floor planning, and urban block layouts. By including these baselines across different model capacities, we can evaluate how GenFlow improves over strong planning-augmented and zero-shot generation methods in both temporal and spatial reasoning tasks.

# F  EVALUATION DETAILS

For each constraint type, we compute accuracy as the fraction of prompts for which the model output satisfies the constraint:

**(a) Single (Once) constraints:** checks adherence to individual constraints.

$$\text{Acc}_{\text{once}} = \frac{\sum_{i=1}^{N_{\text{once}}} R_i}{N_{\text{once}}}$$

**(b) Range constraints:** checks adherence to constraints that span multiple blocks or a specified range.

$$\text{Acc}_{\text{range}} = \frac{\sum_{i=1}^{N_{\text{range}}} R_i}{N_{\text{range}}}$$

**(c) Periodic constraints:** checks adherence to recurring constraints across periodic intervals.

$$\text{Acc}_{\text{periodic}} = \frac{\sum_{i=1}^{N_{\text{periodic}}} R_i}{N_{\text{periodic}}}$$

Here, $R_i = 1$ if the model's output satisfies the corresponding constraint and $R_i = 0$ otherwise; $N_{\text{once}}, N_{\text{range}}, N_{\text{periodic}}$ denote the total number of prompts for each constraint type.

**Average Accuracy (Avg. Acc)** The **Average Accuracy** is defined as the mean of the three type-specific accuracies:

$$\text{Avg. Acc} = \frac{\text{Acc}_{\text{once}} + \text{Acc}_{\text{range}} + \text{Acc}_{\text{periodic}}}{3}$$

This provides an overall measure of the model's adherence to all constraint types.

# G  IMPLEMENTATION DETAILS

All baselines and GenFlow are implemented based on the LongGenBench framework. Experiments are conducted on **3 NVIDIA A40 GPUs (48 GB each)**, with typical runtime reported for different model scales (0.5B / 1.5B / 7B) and average GPU memory usage during training and inference.

**Training Hyper-parameters.**  We use the AdamW optimizer with learning rate, weight decay, warmup steps, and LR decay schedules specified per model. Per-GPU batch size, global batch size, and gradient accumulation steps are recorded, with total training steps/epochs reported for both planning and generation stages. Gradient clipping is applied when necessary.

**Model Fine-tuning.**  Models are trained in bf16 precision, with LoRA adapters applied when applicable (rank, scaling factor, target modules). Maximum sequence length, tokenizer version, and vocabulary are specified to ensure reproducibility.

**Dataset and Evaluation.**  Exact training, validation, and test splits are provided. All experiments use fixed random seeds, and evaluation frequency, checkpointing strategy, and metric computation scripts are clearly defined. Results are reproducible using the same seeds and sampling parameters as in LongGenBench.

These additions provide complete details on hardware, hyper-parameters, model fine-tuning, and evaluation protocol, enabling precise replication of our results.

## H    LIMITATIONS AND FUTURE WORK

While GenFlow demonstrates strong performance in constrained long-form text generation, it is not without limitations. First, the hierarchical planning and reward-guided optimization processes introduce additional computational overhead, resulting in longer generation times compared to single-pass or naive generation methods. This increased latency may limit real-time applications or scenarios requiring rapid content generation. Second, although GenFlow effectively handles diverse constraints and maintains structural coherence, extremely complex tasks with highly interdependent constraints may still challenge the planning and filtering modules, occasionally leading to suboptimal sub-plan selection. Third, our current evaluation focuses on four representative long-form scenarios; generalization to other domains, such as multi-modal content or highly specialized professional texts, remains to be investigated.

For future work, we aim to explore strategies to reduce computational cost without sacrificing output quality, such as model distillation, parallelized sub-plan generation, or more efficient reward evaluation mechanisms. Additionally, integrating more advanced constraint representation and reasoning techniques could further improve performance on highly interdependent or domain-specific tasks. Finally, extending GenFlow to multi-modal long-form generation and interactive adaptive workflows may broaden its applicability and provide more flexible solutions for real-world content creation.

