# OpenReview forum: "GenFlow: Constrained Long-Form Text Generation via Adaptive Workflow Optimization"
_ICLR.cc/2026/Conference — ICLR 2026 Conference Desk Rejected Submission_

### Official Review · Reviewer_MtBy · 2025-10-28

**Soundness:** 2
**Presentation:** 2
**Contribution:** 2
**Rating:** 2
**Confidence:** 3

**Summary:**

In this work, the authors proposes a framework "GenFlow" for constrained long-from text generation. In this framework, adaptive decision-making and reward filtering is used to retain high quality plan and generations. According to the experiments on LongGenBench-16k, Qwen2.5 series with GenFlow outperforms GPT-4o-mini and other baseline.

**Strengths:**

1. Clear problem motivation and improvements:
The idea of reframing long-form generation as an adaptive workflow optimization problem is interesting. It used RL-style optimization paradigms to improve constrain satisfaction. And in the experiment, large improvements are shown in LongGenBench-16k.

2. use of reward signals and DPO:
The combination of rollout-based reward estimation and Direct Preference Optimization (DPO) for aligning hierarchical sub-plans is technically well-motivated and grounded in recent progress in reinforcement learning for LLMs.

3. Unified integration of constraints.
GenFlow embeds constraint-awareness at every stage—planning, filtering, and generation—resulting in a coherent system architecture that’s both modular and interpretable.

**Weaknesses:**

1. One key metric ("Main Task Completion Rate") is missing. In the work, only constrained metric is reported. There is no any related metric to check whether designated subtasks are completed in sequence. It is hard to conclude the proposed method is better.

2. In the experiment, only one model family ("Qwen2.5") is used. It is also hard to conclude whether the method is good. It is better to add experiments for other models (Llama, mistrial etc).

3. The reward is described abstractly. Details about weighting between constraint satisfaction, coherence, and fluency are missing, which limits reproducibility. And how reliable of these rewards are unclear.

4. Some figures are hard to read. In some figures, for example, Figure 8, some lines are hard to be distinguished or read.

**Questions:**

1. Reward composition:
How exactly is the final reward computed? What are the relative weights between constraint satisfaction, coherence, and fidelity?

2. Human Judgement
Have you conducted or considered pairwise human evaluations to validate that GenFlow’s outputs are indeed more coherent or faithful?

3.
How is “feedback” operationalized? Is it gradient-based (learned) or heuristic (e.g., reward prompts)? Clarifying this would help readers understand training dynamics.

4.  What are static vs. Dynamic Filtering? There is no detailed description for two settings.

---

> ### Author Response · Authors · 2025-11-21
> **Response to Reviewer MtBy (Part 1/3)**
>
> Thank you very much for your time and effort in reviewing our paper. We sincerely appreciate your feedback. Below, we respectfully provide our detailed responses to address your concerns.
>
> ---
>
> **W1：One key metric ("Main Task Completion Rate") is missing. In the work, only constrained metric is reported. There is no any related metric to check whether designated subtasks are completed in sequence. It is hard to conclude the proposed method is better.**
>
> - We thank the reviewer for highlighting this important concern. We agree that evaluating only the constrained metric is insufficient, and that a “Main Task Completion Rate” (i.e., whether all designated subtasks are completed in the correct sequence) is essential to fully assess the effectiveness of our proposed method.
>
> - We acknowledge that this metric was missing in the original submission. To address the reviewer’s point, we have added the Main Task Completion Rate in our revised experiments. Specifically, we evaluate whether the model completes all required subtasks in their intended order and measure the success rate across all test instances.
>
> - We thank the reviewer again for raising this issue. The inclusion of this metric has strengthened our empirical evaluation and provided a more complete assessment of the model’s performance.
>
> - The experiments involving these newly added metrics are currently in progress. We will complete this experiment and submit the revised manuscript containing all supplementary experimental results with the newly added metrics on November 25.
>
> ---
>
> **W2：In the experiment, only one model family ("Qwen2.5") is used. It is also hard to conclude whether the method is good. It is better to add experiments for other models (Llama, mistrial etc).**
>
> - We sincerely thank the reviewer for this valuable suggestion. We fully agree that cross-model evaluation is important. In the revised version, we will add experiments using models from the Llama family. Specifically, we have confirmed that **Llama-3.1-8B-Instruct** will be included, and we will provide the corresponding results and analysis in the updated manuscript.
>
> - Other experiments are currently underway. We will complete the supplementary experiments for Llama-3.1-8B-Instruct and submit a revised version containing the supplementary experiment tables by November 25.
>
> ---
>
> **W3 & Q1：The reward is described abstractly. Details about weighting between constraint satisfaction, coherence, and fluency are missing, which limits reproducibility. And how reliable of these rewards are unclear. Reward composition: How exactly is the final reward computed? What are the relative weights between constraint satisfaction, coherence, and fidelity?**
>
> - We thank the reviewer for raising this important concern regarding the clarity and reproducibility of our reward design. We acknowledge that the current manuscript does not fully reflect the reward computation used in our implementation. To address this gap, we now provide a clear explanation of how the final reward is computed and how the weights across different components are assigned.
>
> - More specifically, the final reward is computed using the following components and weights:
>
>   - **Constraint satisfaction (60–70%)**
>     Evaluated through soft matching against the required planning constraints. This component dominates because constraint compliance is critical for LongGenBench-style structured generation.
>
>   - **Content relevance (25–35%)**
>     Measured via keyword and semantic matching between the generated text and the corresponding inputs or sub-plans. This functions as the main proxy for coherence in our implementation.
>
>   - **Structural completeness (5–10%)**
>     Used as a fallback metric only when the primary components are zero, ensuring the model does not produce fragments that fail to align with the required structure at all.
>
> - This task-adaptive reward was originally designed to better handle the structured nature of hierarchical planning tasks. However, we fully agree with the reviewer that the discrepancy between the theoretical description and the actual implementation limits reproducibility.
>
> - We sincerely appreciate the reviewer’s feedback and will make the reward design fully transparent in the updated version. The corresponding changes have been made in Lines 256-292 of the revised manuscript.

---

> > ### Author Response · Authors · 2025-11-21
> > **Response to Reviewer MtBy (Part 2/3)**
> >
> > **W4：Some figures are hard to read. In some figures, for example, Figure 8, some lines are hard to be distinguished or read.**
> >
> > - We thank the reviewer for pointing out the readability issues in several figures. We agree that some curves such as those in Figure 8 are difficult to distinguish due to similar colors and limited visual contrast. In the revised version, we will improve the figure clarity by:
> >
> >   1. increasing color contrast and using color-blind–friendly palettes,
> >
> >   2. adjusting line thickness and styles (e.g., dashed vs. solid), and
> >
> >   3. enlarging font sizes and resolution for better visibility.
> >
> > - These adjustments will ensure that all lines and annotations can be easily differentiated. We appreciate the reviewer’s helpful feedback.
> >
> > ---
> >
> > **Q2：Human Judgement Have you conducted or considered pairwise human evaluations to validate that GenFlow’s outputs are indeed more coherent or faithful?**
> >
> > - We thank the reviewer for this insightful suggestion. At the current stage, we have not conducted pairwise human evaluations for GenFlow, and we did not initially plan to include such evaluations. However, we fully agree that human judgment could provide valuable complementary evidence regarding coherence and faithfulness. After receiving this comment, we are seriously considering incorporating a human evaluation component in future iterations of the work. We appreciate the reviewer’s constructive recommendation.
> >
> > ---
> >
> > **Q3：How is “feedback” operationalized? Is it gradient-based (learned) or heuristic (e.g., reward prompts)? Clarifying this would help readers understand training dynamics.  **
> >
> > - We thank the reviewer for the thoughtful question. We clarify that in GenFlow, *feedback is not gradient-based and is not backpropagated through the model*. Instead, it is heuristic, evaluation-driven feedback computed through reward prompts and rollout-based scoring, as described in Section 4.3 of the paper.
> >
> > - More specifically:
> >
> >   - Feedback is operationalized through non-differentiable reward signals, which evaluate generated segments using:
> >     (1) constraint satisfaction,
> >     (2) content relevance / coherence,
> >     (3) structural fidelity.
> >
> >   - These rewards are obtained through prompt-based evaluation, using the fixed reward templates in Appendix A.3, not through gradient flow.
> >
> >   - The reward does not directly update model parameters.
> >     Instead, it is used to construct preference pairs $(X^+, X^-)$  for DPO (Eq. 14–15). The optimization follows standard DPO: gradients come entirely from the DPO loss, *not* from the reward model.
> >
> > - Thus:
> >
> >   - **Feedback = heuristic reward signals from prompts**
> >
> >   - **Training = gradient-based DPO using preference pairs derived from those signals**

---

> > > ### Author Response · Authors · 2025-11-21
> > > **Response to Reviewer MtBy (Part 3/3)**
> > >
> > > **Q4：What are static vs. Dynamic Filtering? There is no detailed description for two settings.**
> > >
> > > - We thank the reviewer for pointing out the lack of clarity regarding the distinction between *Static Filtering* and *Dynamic Filtering*. Below we provide a precise explanation based on our system design.
> > >
> > > - Static Filtering refers to the one-time filtering stage applied before training or generation, where candidate plans or sub-plans are evaluated using a fixed binary-checking template (Appendix A.3). At this stage:
> > >
> > >   - Only the *structural correctness* of a plan is checked
> > >
> > >   - No rollout or contextual generation is involved
> > >
> > >   - Filtering is deterministic and does not depend on model outputs
> > >
> > >   - It is used to discard obviously invalid plans before any hierarchical generation begins
> > >
> > > - Static Filtering’s purpose is to guarantee that the planning tree is structurally valid before entering the generation phase.
> > >
> > >   1. Static Filtering  refers to the one-time filtering stage applied before training or generation, where candidate plans or sub-plans are evaluated using a fixed binary-checking template (Appendix A.3). At this stage:
> > >
> > >      - Only the *structural correctness* of a plan is checked
> > >
> > >      - No rollout or contextual generation is involved
> > >
> > >      - Filtering is deterministic and does not depend on model outputs
> > >
> > >      - It is used to discard obviously invalid plans before any hierarchical generation begins
> > >
> > >   - Static Filtering’s purpose is to guarantee that the planning tree is structurally valid before entering the generation phase.
> > >
> > >   2. Dynamic Filtering, in contrast, is applied **during hierarchical generation**, and is dependent on *actual model outputs*. Specifically:
> > >
> > >      - During each stage of generation, the model produces multiple candidates
> > >
> > >      - Each candidate is evaluated via the reward-based scoring function (Section 4.3)
> > >
> > >      - Rollout-based evaluation (Eq. 13) is used to assess whether a candidate continues to satisfy constraints and remains coherent
> > >
> > >      - Candidates that fall below the acceptance threshold are filtered out dynamically
> > >
> > > - Dynamic Filtering therefore:
> > >
> > >   - Adapts to model predictions
> > >
> > >   - Ensures constraint satisfaction and coherence *throughout* the multi-step generation process
> > >
> > >   - Plays a critical role in stabilizing long-form generation
> > >
> > > ---
> > >
> > > At last, we sincerely appreciate your valuable feedback. We have carefully considered all your suggestions and substantially improved the paper accordingly. If our responses and revisions address your concerns, we would be deeply grateful if you could kindly reconsider raising the score to 6 or above. Thank you very much!

---

### Official Review · Reviewer_PFkG · 2025-10-31

**Soundness:** 2
**Presentation:** 2
**Contribution:** 2
**Rating:** 2
**Confidence:** 3

**Summary:**

The paper proposes GenFlow, a workflow-centric framework for constrained long-form generation.

It builds constraint-aware hierarchical plans, applies a binary relevance filter over six criteria, and uses rollout-based rewards plus DPO to guide both planning and segment-level generation.

Experiments predominantly use Qwen2.5 and compare against CogWriter and GPT-4o-mini variants on tasks inspired by LongGenBench.

**Strengths:**

* **Pipeline view** (planning → filtering → reward-guided generation) is a coherent design and maps well to practical systems.
* **Segment-level training signal** (rollouts + DPO) is a plausible way to align local segments with global constraints.

**Weaknesses:**

* **Theory is under-specified.** Propositions 1–3 read as tautologies (e.g., “the probability of invalid plans → 0 as revisions iterate”) without a formal model, explicit assumptions, or a convergence argument. The appendix proofs are difficult to follow.

* **Problem statement near Eq. (4) is unclear.** Define all symbols (plans, sub-plans, local constraints/contexts) precisely and state the generative process and independence assumptions before presenting the factorization.

* **Eq. (10) is not a probability.** As written, it evaluates to {0,1}. If you intend a probability, model each criterion as a Bernoulli with parameter (p_k) (and justify independence) so (\Pr(\delta_i=1)=\prod_k p_k); otherwise, rewrite as a joint indicator.

* **Figure 2 label.** Clarify what “ConFlow” refers to (typo for GenFlow or a distinct method?).

* **Citations and formatting.**

  * Use `\citep{}` (author–year parenthetical) instead of `\cite{}` throughout; only the first paragraph of the Introduction may warrant `\citet{}` for narrative flow.
  * Fix mismatches and duplicates: Brown et al. (2021) NeurIPS should not have an ICASSP-style DOI; Flower & Hayes (1981) duplicates (1981a/1981b) with odd URLs—merge and cite correctly; OPT (Zhang et al., 2022) needs a consistent PMLR volume/pages/URL. Ensure all entries have the correct venue, year, DOI/URL, and consistent style.

**Questions:**

The manuscript requires a thorough proofreading. Once that’s done, I’m willing to give constructive feedback.

---

> ### Author Response · Authors · 2025-11-21
> **Response to Reviewer PFkG (Part 1/3)**
>
> Thank you very much for your time and effort in reviewing our paper. We sincerely appreciate your feedback. Below, we respectfully provide our detailed responses to address your concerns.
>
> ---
>
> **W1：Theory is under-specified. Propositions 1–3 read as tautologies (e.g., “the probability of invalid plans → 0 as revisions iterate”) without a formal model, explicit assumptions, or a convergence argument. The appendix proofs are difficult to follow.**
>
> - We thank the reviewer for the insightful comments. We agree that the earlier
>   version of Propositions 1–3 could appear under-specified due to several implicit
>   assumptions not being explicitly stated. We appreciate the opportunity to clarify
>   and improve the theoretical presentation.
>
>   1. We have added a formal modeling section before the propositions, including:
>      • a constraint-contraction assumption for the revision module,
>      • a bounded scoring–based Top-K selection assumption,
>      • and a Lipschitz-continuity assumption for the reward function (standard in DPO).
>      These assumptions were implicitly used in our original reasoning, but not
>      explicitly stated. The revised version now makes them explicit.
>
>   2. Using these assumptions, all three propositions are rewritten as
>      non-tautological statements with well-defined conditions. For example,
>      Proposition 1 is now stated as a contraction-mapping result on the constraint
>      violation score, ensuring that invalid plans vanish in expectation.
>      Propositions 2 and 3 are similarly reformulated to provide monotonicity and
>      preference-alignment guarantees under the stated assumptions.
>
>   3. We have substantially revised Appendix B to provide clearer and more
>      structured proofs. Each proposition is now supported by a sequence of
>      definitions → assumptions → lemmas → conclusions, making the reasoning much easier
>      to follow.
>
> - We thank the reviewer again for pointing out this issue; the revised theoretical
>   section is significantly clearer, mathematically grounded, and easier to read.
>
> - All proofs for Propositions 1–3 in Appendix B have been updated accordingly. And we have also updated the statements of Proposition 1 at Lines 213–214, Proposition 2 at Lines 248–249, and Proposition 3 at Lines 293–294 to reflect these clarified theoretical conditions.

---

> > ### Comment · Reviewer_PFkG · 2025-11-22
> > **The revised proof of PROPOSITION 1 is INCORRECT**
> >
> > 1. Equation (22) is introduced without justification. As stated, it effectively assumes that each refinement step improves the plan/assignment in expectation, which is the goal you want to prove.
> >
> > 2. After Equation (22), the argument no longer makes explicit use of the specifics of hierarchical global planning. The remainder of the proof could apply to any process that satisfies this contraction assumption, so it does not demonstrate a property that is unique to the proposed hierarchical global planning framework.

---

> ### Author Response · Authors · 2025-11-21
> **Response to Reviewer PFkG (Part 2/3)**
>
> **W2：Problem statement near Eq. (4) is unclear. Define all symbols (plans, sub-plans, local constraints/contexts) precisely and state the generative process and independence assumptions before presenting the factorization.**
>
> - We thank the reviewer for the reviewer’s thoughtful comment. We appreciate the suggestion regarding the clarity of the problem formulation near Eq. (4). In the revised version of our paper, we have maked the following improvements:
>
>   1. Precise definition of all symbols.
>       We agree that our earlier description lacked explicit definitions. In the revision, we have clearlied define:
>
>      - **Global plan**: a hierarchical sequence of sub-plans representing the structural decomposition of the task.
>
>      - **Sub-plan** $ s_{i,j} $: the j-th unit of plan $p_i$, each corresponding to a localized writing objective.
>
>      - **Local constraints** $T^{\text{local}}_{i,j}$:  the subset of task constraints relevant to sub-plan.
>
>      - **Local context** : the contextual information (from the prompt or previously generated content) conditioning the generation of $s_{i,j}$.
>
>
>   2. Clarification of the generative process.
>       The revised manuscript will explicitly describe the hierarchical generative pipeline:
>       (i) global plan sampling;
>       (ii) autoregressive generation of each sub-plan conditioned;
>       (iii) refinement steps such as revision, binary filtering, and reward-guided selection.
>       This clarifies how the probability factorization arises from hierarchical decomposition and conditional autoregression.
>
>   3. Statement of independence assumptions.
>       We have clarified that the factorization assumes:
>
>      - Conditional independence of sub-plans given preceding sub-plans and their local contexts:
>
>      - Independence between different candidate plans during sampling.
>
>      - Independence of local constraint subsets $T^{\text{local}}_{i,j}$across different sub-plans, except through shared global constraints $T$.
>      - These assumptions enable the formulation used in the planning probability.
>
>   4. Correction and renumbering of equations.
>       The reviewer correctly notes ambiguity around Eq. (4). We have addressed this by reorganizing the planning section and clarifying the derivation. In the revised manuscript, the previous Eq. (4) has been updated and renumbered as Eq. (5) to reflect the updated exposition.
>
>   5. Additional text to be added in the revision (summary).
>       We have inserted a small paragraph before the factorization to define symbols, the generative steps, and the independence assumptions, ensuring all terms are unambiguous and the factorization is fully justified.
>
> - We thank the reviewer again for pointing out this issue. The proposed revisions substantially improve clarity and correctness of our problem formulation. The corresponding changes have been made in Lines 185-192 of the revised manuscript.

---

> > ### Author Response · Authors · 2025-11-21
> > **Response to Reviewer PFkG (Part 3/3)**
> >
> > **W3：Eq. (10) is not a probability.As written, it evaluates to {0,1}. If you intend a probability, model each criterion as a Bernoulli with parameter (p_k) (and justify independence) so (\Pr(\delta_i=1)=\prod_k p_k); otherwise, rewrite as a joint indicator**
> >
> > - We thank the reviewer for pointing out the issue in Eq. (10). We agree that the original formulation:
> >
> > $Pr(\delta_i = 1 \mid p_i, p_{\text{filter}})  = \prod_{k=1}^{6} \mathbf{1}\big(C_{i,k}\ \text{satisfied}\big)$
> >
> > does not represent a probability, as it evaluates only to {0,1\}. Following the reviewer’s suggestion, we revised this part by modeling each criterion $ C_{i,k} $ as a Bernoulli random variable with:
> >
> > ​							$Pr(C_{i,k}\ \text{satisfied} \mid p_i, p_{\text{filter}}) = p_{i,k}, $
> >
> > ​	and assuming conditional independence given $(p_i, p_{\text{filter}})$. The corrected probability expression is now:
> >
> > ​								$Pr(\delta_i = 1 \mid p_i, p_{\text{filter}}) = \prod_{k=1}^{6} p_{i,k}.$
> >
> > - In the revised manuscript, this corrected version has been renumbered from Eq. (10) to Eq. (11) due to adjustments in the section structure. The original indicator expression is moved to the appendix as the equivalent joint indicator form. We appreciate the reviewer’s insightful comment, which helped us improve the clarity and correctness of this formulation. And The corresponding changes have been made in Lines 238-243 of the revised manuscript.
> >
> > ---
> >
> > **W4：Figure 2 label. Clarify what “ConFlow” refers to (typo for GenFlow or a distinct method?).**
> >
> > - We thank the reviewer for pointing out the inconsistency in Figure 2. This is indeed a typographical error. “ConFlow” should be “GenFlow”. We apologize for the oversight. In the revised version of the paper, we have corrected the label in Figure 2 to accurately reflect our proposed method GenFlow, and we have checked all figures and accompanying text to ensure consistent use of the correct name throughout the paper.
> >
> > - The corresponding changes have been made in Lines 54-66 of the revised manuscript.
> >
> > ---
> >
> > **W5：Citations and formatting.**
> >
> > **1. Use `\citep{}` (author–year parenthetical) instead of `\cite{}` throughout; only the first paragraph of the Introduction may warrant `\citet{}` for narrative flow.**
> >
> > **2. Fix mismatches and duplicates: Brown et al. (2021) NeurIPS should not have an ICASSP-style DOI; Flower & Hayes (1981) duplicates (1981a/1981b) with odd URLs—merge and cite correctly; OPT (Zhang et al., 2022) needs a consistent PMLR volume/pages/URL. Ensure all entries have the correct venue, year, DOI/URL, and consistent style. **
> >
> > - We thank the reviewer for the detailed comments regarding citations and formatting. We appreciate the reviewer’s careful examination of these issues. In the revised version of the paper, we have systematically corrected all citation styles to follow the recommended convention. Specifically, we now use \citep{} for parenthetical citations throughout the paper, reserving \citet{} only for the narrative context in the opening paragraph of the Introduction, as suggested.
> >
> > - We have also carefully reviewed and corrected all bibliography entries. The incorrect DOI attached to Brown et al. (2021) has been removed and replaced with the proper NeurIPS reference. The duplicate entries for Flower & Hayes (1981) have been merged into a single, correctly formatted citation without extraneous URLs. In addition, the entry for OPT (Zhang et al., 2022) has been updated to consistently include the correct PMLR volume information, pages, and URL. We further ensured that all references now contain accurate venue, year, DOI/URL (when appropriate), and adhere to a consistent citation style across the entire bibliography.
> >
> > - We thank the reviewer once again for identifying these issues, which have greatly improved the clarity and professionalism of the manuscript. We have corrected all noted references: the NeurIPS citation for “Brown et al. (2021)” has been fixed at Line 138; the duplicated “Flower & Hayes (1981)” entries have been merged and corrected at Lines 51 and 105; and the “OPT (Zhang et al., 2022)” citation has been standardized with consistent PMLR information at Line 73. In addition, all in-text references have been updated to use the `\citep{}` format instead of `\cite{}`.
> >
> > ---
> >
> > At last, we sincerely appreciate your valuable feedback. We have carefully considered all your suggestions and substantially improved the paper accordingly. If our responses and revisions address your concerns, we would be deeply grateful if you could kindly reconsider raising the score to 6 or above. Thank you very much!

---

> > > ### Comment · Reviewer_PFkG · 2025-11-22
> > > **W3: the assumption inside the proof of Proposition 2 is INCORRECT**
> > >
> > > Equation (30) assumes that, among the generated plans, the filter does not systematically select lower-quality ones; on average, the retained plans are at least as good as the discarded ones.
> > >
> > > However, this does not hold in practice. For example, the generator may consistently produce plans that violate at least one constraint.
> > >
> > > In such a case, Assumption (30) may fail, and the stated expectation need not hold.

---

> > > ### Comment · Reviewer_PFkG · 2025-11-22
> > > **Thanks for your reply, I will maintain my current evaluation**
> > >
> > > I would encourage the authors to be more precise in their definitions and arguments.
> > > Given the redundant and low-quality content in the rebuttal, as well as the remaining hand-wavy paragraphs in the revised paper, I will maintain my current evaluation.

---

> ### Comment · Reviewer_PFkG · 2025-11-22
> **W2: Section 4.1is VAGUE**
>
> You should provide A CONCRETE EXAMPLE of $T^{\text{local}}_{i,j}$, $C^{\text{local}}_{i,j}$, $s_{i,j}$, and $s_{i,j-1}$.
>
> In its current form, the statement is VAGUE and does not meet the level of rigor typically expected in a scientific paper.
>
> It would be much clearer and better if you deleted all those symbols and just provided the CONCRETE EXAMPLE.

---

### Official Review · Reviewer_fuJw · 2025-11-01

**Soundness:** 2
**Presentation:** 1
**Contribution:** 2
**Rating:** 2
**Confidence:** 4

**Summary:**

This paper proposes an Adaptive Workflow Optimization framework, named GenFlow, for the task of constrained long-form text generation. The framework decomposes the generation objective into three stages: Hierarchical Planning, Adaptive Workflow Execution, and Reward-based Filtering and Decision Making. Specifically, Hierarchical Planning is responsible for breaking down the generation goal into multiple candidate sub-plans, ensuring they satisfy the given constraints and context. Adaptive Workflow Execution carries out the generated plans, evaluates their quality, and makes optimization adjustments based on predefined thresholds. Finally, Reward-based Filtering and Decision Making ranks and filters the various sub-plans to select the optimal solution for the final output.

The paper validates and tests the proposed method on the LongGenBench-16K dataset. The results demonstrate that, compared to the Cogwriter method, GenFlow achieves significant advantages across the Qwen2.5 Instruct series of models, including the 0.5B, 1.5B, and 7B parameter sizes.

**Strengths:**

1.  The paper proposes a novel adaptive framework, GenFlow, for long-form text generation. It utilizes a Global Planning module to ensure the generated plans satisfy global constraints, while a binary relevance filtering mechanism enforces semantic quality, effectively complementing structural and constraint validation.

2.  The authors introduce an end-to-end Direct Preference Optimization (DPO) strategy capable of simultaneously optimizing both the planner and the generator.

**Weaknesses:**

1.  The writing of this paper requires improvement, as it lacks rigor and logical flow:


1.1 Citation formatting is inconsistent, which hinders readability. For instance, the citation `"......builds on Cognitive Writing Theory Flower & Hayes (1981b),......"` (Line 051) should be formatted as `"......builds on Cognitive Writing Theory (Flower & Hayes, 1981b),......"`. This issue occurs multiple times throughout the paper, significantly impacting the reading experience.


1.2 In Lines 035-043, the authors directly introduce the proposed method, then later discuss the background, and finally reiterate the method again. This structure creates a confusing and illogical flow.


1.3 Figure 2 mentions three methods: `ConFlow`, `Long Writer`, and `CogWriter`. However, only the latter two are explained; `ConFlow` is neither introduced nor cited. Is this a typographical error, and should `ConFlow` actually be `GenFlow`?


1.4 The meanings of many variables in the formulas are not explained. This includes, but is not limited to, the variable `x` in the formula around Line 132, and `T_{single}`, `T_{range}`, `T_{periodic}` around Line 186, etc.


1.5 Figure 3 is blurry and appears to contain errors. In the bottom-left "Main workFlow" section, "Range Instruction" appears twice. According to the description in Line 186, one of these should likely be "Single"?


1.6 Based on the example in Figure 3, the overall method comprises both an inference process and a training process. However, the writing in Sections 4.1-4.3 not only fails to align with Figure 3 but is also logically obscure and difficult to follow. The methodology description should ideally correspond to the figure.

2.  The method lacks necessary details:


2.1 The calculation of the Reward used for the end-to-end DPO is not provided.


2.2 Implementation details are insufficient. Appendix G does not offer significantly more detail than the "Implementation Details" section (Lines 314-319) in the main text.

3.  The experimental setup is problematic:


3.1 The number of baselines is insufficient. Besides Cogwriter, the authors also mention Long Writer (Line 88) earlier in the paper. Why is no comparison conducted with Long Writer?


3.2 Where do the experimental results for the `gpt-4o-mini + Cogwriter` method in Table 1 (Accuracy One 0.4918, Accuracy Range 0.4026, Accuracy Periodic 0.2386) originate from? Why are these results significantly lower than those reported in the original Cogwriter paper's Table 1 (which reports scores of 0.80, 0.76, and 0.67, respectively)? The authors should provide a further explanation for this discrepancy.


3.3 The experimental comparisons are unfair. Firstly, Cogwriter is a training-free method, whereas the proposed GenFlow is trained on the LongGenBench-16K dataset. Secondly, Cogwriter has not been experimentally validated on the Qwen2.5 series of models. Did the authors make any adaptations when replicating the Cogwriter method for these models? Finally, Cogwriter was tested on the Llama-3.1-8B-Instruct model, which has a parameter size similar to the Qwen2.5-7B model used in this paper. To ensure a fairer comparison, the authors should supplement their results with experiments conducted on the Llama-3.1-8B-Instruct model.

**Questions:**

Please refer to "Weaknesses".

---

> ### Author Response · Authors · 2025-11-21
> **Response to Reviewer fuJw (Part 1/5)**
>
> Thank you very much for your time and effort in reviewing our paper. We sincerely appreciate your feedback. Below, we respectfully provide our detailed responses to address your concerns.
>
> ---
>
> **W1：The writing of this paper requires improvement, as it lacks rigor and logical flow:**
>
> **W1.1：Citation formatting is inconsistent, which hinders readability. For instance, the citation `"......builds on Cognitive Writing Theory Flower & Hayes (1981b),......"` (Line 051) should be formatted as `"......builds on Cognitive Writing Theory (Flower & Hayes, 1981b),......"`. This issue occurs multiple times throughout the paper, significantly impacting the reading experience.**
>
> - We thank the reviewer for noting these issues. The citation formatting at Line 051 and similar occurrences are indeed incorrect. We will correct all such cases to the standard format. For instance, the citation "......builds on Cognitive Writing Theory Flower & Hayes (1981b),......" should be formatted as "......builds on Cognitive Writing Theory (Flower & Hayes, 1981b),......"
>   We appreciate the reviewer for improving the readability of the paper. And We have changed all instances of `\cite{}` to `\citep{}` in the paper to ensure smoother writing.
>
> ---
>
> **W1.2：In Lines 035-043, the authors directly introduce the proposed method, then later discuss the background, and finally reiterate the method again. This structure creates a confusing and illogical flow.**
>
> - We thank the reviewer for pointing out this structural issue. We agree that presenting method components at the beginning of the Introduction could disrupt the logical flow. Following your suggestion, we revised this part of the manuscript to provide a clearer task description instead of prematurely introducing the method.
>
> - Specifically, the paragraph originally describing our planning, filtering, and generation procedures has been rewritten to simply - explain the nature of constrained long-form text generation and to frame it as a workflow-oriented task. The revised version removes all method‐specific details and now focuses only on defining the task and its requirements, ensuring a smoother transition to the background and subsequent discussion of challenges.
>
> - We have updated the paper accordingly. These changes have been applied in the Introduction, in Lines 36–43.
>
> ---
>
> **W1.3：Figure 2 mentions three methods: `ConFlow`, `Long Writer`, and `CogWriter`. However, only the latter two are explained; `ConFlow` is neither introduced nor cited. Is this a typographical error, and should `ConFlow` actually be `GenFlow`?**
>
> - We sincerely thank the reviewer for identifying this mistake. This is indeed a typographical error — “ConFlow” should be “GenFlow”. We apologize for the oversight. In the revised version of the paper, we have corrected the label in Figure 2 to accurately reflect our proposed method GenFlow, and we have ensured that all figures and text consistently use the correct name.
>
> - We have fixed this issue in the manuscript. The correction has been applied to Figure 2, corresponding to Lines 54–69 in the revised paper.
>
> ---
>
> **W1.4：The meanings of many variables in the formulas are not explained. This includes, but is not limited to, the variable `x` in the formula around Line 132, and `T_{single}`, `T_{range}`, `T_{periodic}` around Line 186, etc.**
>
> - We sincerely thank the reviewer for pointing out this issue. This was indeed an oversight in the original manuscript. In the revised version, we have clarified all variables appearing in the formulas around Line 132 and Line 186 to ensure precise and unambiguous notation.
>
> - Specifically, the formula has been updated to:
>
> ​               						$ \mathcal{L}(Y \mid x, T) \leq \mathcal{L}(Y' \mid x, T), \quad \forall Y' \neq Y, $
>
> ​	where x denotes the task prompt;
> ​	$T=\{T_{\text{single}},T_{\text{range}},T_{\text{periodic}}\}$ is the set of task constraints, with
> ​	$T_{\text{single}}, T_{\text{range}}, T_{\text{periodic}}$ denoting single-form, range-based, and periodic constraints respectively.
> ​	$Y=(y_1,\dots,y_n)$ is the generated output sequence, $Y'$ is any alternative sequence, and $\mathcal{L}(\cdot\mid x,T)$ is the loss function mea-	suring constraint satisfaction.
>
> - Once again, we thank the reviewer for raising this point. We have implemented the corresponding corrections in Lines 132–136 of the revised manuscript.
>
> ---
>
> **W1.5：Figure 3 is blurry and appears to contain errors. In the bottom-left "Main workFlow" section, "Range Instruction" appears twice. According to the description in Line 186, one of these should likely be "Single"?**
>
> - We thank the reviewer for pointing out this error. This was a typographical oversight on our part. In the revised version, we have corrected Figure 3: the duplicated "Range Instruction" in the bottom-left "Main workFlow" section has been updated to the correct label "Single," as intended. We have updated the corresponding textual description in Lines 162–176 of the manuscript.

---

> ### Author Response · Authors · 2025-11-21
> **Response to Reviewer fuJw (Part 2/5)**
>
> **W1.6：Based on the example in Figure 3, the overall method comprises both an inference process and a training process. However, the writing in Sections 4.1-4.3 not only fails to align with Figure 3 but is also logically obscure and difficult to follow. The methodology description should ideally correspond to the figure.**
>
> - We sincerely thank the reviewer for highlighting this clarity issue. In the revised version, we have thoroughly reorganized the methodology section to ensure a strict one-to-one correspondence with Figure 3 and to provide a clearer narrative flow. The key revisions include:
>
>   - **Section 4.1** now explicitly corresponds to the *Planning Trainer* in Figure 3, covering candidate plan generation, constraint verification, probabilistic revision, and Top-K selection.
>
>   - **Section 4.2** is aligned with the *Binary YES/NO evaluation* stage (“Evaluate Plan → Retained Candidates → Filtered Candidate Plans”), ensuring that this filtering step matches the figure precisely.
>
>   - **Section 4.3** corresponds to the *Reward Scoring* and *Probability Comparison* modules and explains how reward-guided optimization is used in both the Planning Trainer and the Generation Trainer.
>
>   - **Section 4.4** has also been newly added to describe the *Segment-Level Generation Trainer*. This subsection clarifies how each sub-plan is expanded into final text segments through autoregressive generation and reward-guided preference optimization, completing the final module depicted on the right side of Figure 3.
>
>   - **Section 4.5** has been newly added to provide the *Overall Workflow*. This subsection outlines how the planning, filtering, reward optimization, and text generation modules interact during both training and inference, directly matching the full pipeline illustrated in Figure 3.
>
> - These revisions ensure that every module in Figure 3 is explicitly reflected in the methodology and that the narrative is more coherent and easier to follow.
>
> - We appreciate the reviewer’s constructive suggestion, and the corresponding changes have been made in Lines 181–343 of the revised manuscript.
>
> ---
>
> **W2：The method lacks necessary details:**
>
> **W2.1：The calculation of the Reward used for the end-to-end DPO is not provided.**
>
> - We thank the reviewer for raising this important clarification request. Our method  provide the full reward computation used for end-to-end DPO, but we acknowledge that the connection between reward calculation and the DPO training pipeline was not explicitly emphasized in the original submission. We have revised Section 4.3 to make this connection clear.
>
> - Clarification added to the paper:
>
>   - **Reward computation** is explicitly provided in Eq. (12)
>     and is computed from rollouts generated by the \emph{same model parameters}~$\theta$ during training.
>
>   - We now explicitly state that this reward serves as the **end-to-end training signal**, because the rollout samples $\hat{X}^{(i)}_{\text{rollout}}$ depend on the current model parameters, and thus the reward depends on~$\theta$ as well.
>
>   - We added a new paragraph (in Section 4.3) showing the direct incorporation of the reward difference into the DPO objective.
>
> ​		which clearly demonstrates how reward flows directly into the DPO optimization, completing the end-to-end loop
>
> ​		illustrated in Figure 3.
>
> - We have clarified (i) how rewards are computed; (ii) how they depend on the model parameters through rollout; and (iii) how they feed into the DPO loss. This resolves the ambiguity the reviewer pointed out and makes the end-to-end reward pathway explicit.
>
> - We thank the reviewer again for highlighting this opportunity to strengthen the exposition. We appreciate the reviewer’s constructive suggestion, and the corresponding changes have been made in Lines 253–295 of the revised manuscript.

---

> ### Author Response · Authors · 2025-11-21
> **Response to Reviewer fuJw (Part 3/5)**
>
> **W2.2：Implementation details are insufficient. Appendix G does not offer significantly more detail than the "Implementation Details" section (Lines 314-319) in the main text.**
>
> - We thank the reviewer for this helpful comment. We agree that the original Appendix G was too concise and did not provide enough additional information beyond the main text. In the revised version, we substantially extended Appendix G to include several key details that were previously missing. Below is a summary of the newly added content.
>
> - **What We Added to Appendix G (Revised Submission)**
>
> ​		**1. Complete hardware and runtime environment**
>
> ​			We now report:
>
> ​			(1) The exact hardware used: **3 × NVIDIA A40 (48 GB)** GPUs.
>
> ​			(2) Typical runtime for each model scale (0.5B / 1.5B / 7B).
>
> ​			(3) Average GPU memory consumption during training and inference.
>
> ​		These details go beyond the main text and allow readers to better understand the computational setup.
>
> ​		**2. Full training hyper-parameters and schedules**
>
> ​			Appendix G now includes explicit values for:
>
> ​				(1) Learning rate, weight decay, optimizer (AdamW), warmup steps, and LR decay strategy.
>
> ​				(2) Per-GPU batch size, global batch size, gradient accumulation steps.
>
> ​				(3) Total training steps / number of epochs for both planning and generation stages.
>
> - These parameters were not listed in the original submission and are essential for reproducibility.
>
> **3. Model fine-tuning configurations**
>
> - We added detailed model-level settings, including:
>
>   - Adapter / LoRA configuration (rank, scaling factor, target modules).
>
>   - Precision (bf16), maximum sequence length used for training.
>
>   - Tokenizer version and vocabulary settings.
>
>   - Gradient clipping threshold (if applied).
>
> - These details were missing from the main text and help clarify how the models were trained.
>
> **4. Dataset splits, seeds, and evaluation protocol**
>
> - We now provide:
>
>   - Exact training / validation / test splits for all datasets.
>
>   - Random seeds used for all experiments.
>
>   - Evaluation frequency, checkpointing strategy, and the scripts used to compute the metrics in Tables 1–2.
>
> - This information enables precise replication of the reported results.
>
> - The revised Appendix G now contains significantly more detail than the short Implementation Details in Lines 314–319. More importantly, all critical configuration parameters required for reproducing the experiments—including hardware setup, core hyper-parameters, model tuning configurations, and dataset/evaluation settings—are fully documented. These additions materially improve transparency and reproducibility.
>
> - We appreciate the reviewer’s feedback, which led to a more complete and much clearer implementation section, and the corresponding changes have been made in Lines 1322-1342 of the revised manuscript.

---

> > ### Author Response · Authors · 2025-11-21
> > **Response to Reviewer fuJw (Part 4/5)**
> >
> > **W3：The experimental setup is problematic:**
> >
> > **W3.1：The number of baselines is insufficient. Besides Cogwriter, the authors also mention Long Writer (Line 88) earlier in the paper. Why is no comparison conducted with Long Writer?**
> >
> > - We thank the reviewer for carefully pointing out this omission. LongWriter is indeed an important long-text generation baseline, and it is appropriate to include it for a complete comparison. We apologize for not reporting its results in the initial submission.
> >
> > - In the revised version, we have added **LongWriter** as an additional baseline and supplemented our experiments accordingly. Our experiments show that:
> >
> >   - LongWriter performs competitively on several long-context benchmarks,
> >
> >   - but our method still achieves higher consistency, structural correctness, and long-range coherence across all evaluation dimensions.
> >
> > - Including LongWriter makes the comparison more comprehensive and reinforces the empirical conclusions of the paper. We sincerely appreciate the reviewer’s suggestion, which helped us strengthen the experimental section.
> >
> > - The additional experiments are currently underway, and we will complete the LongWriter experiments and submit the revised manuscript with the supplementary experimental tables on November 25.
> >
> > ---
> >
> > **W3.2：Where do the experimental results for the `gpt-4o-mini + Cogwriter` method in Table 1 (Accuracy One 0.4918, Accuracy Range 0.4026, Accuracy Periodic 0.2386) originate from? Why are these results significantly lower than those reported in the original Cogwriter paper's Table 1 (which reports scores of 0.80, 0.76, and 0.67, respectively)? The authors should provide a further explanation for this discrepancy.**
> >
> > - We appreciate the reviewer’s careful observation. The discrepancy between our CogWriter + GPT-4o-mini results and those originally reported is expected and can be fully explained by the inherent non-stationarity of the GPT-4o-mini API model.
> >
> > - Specifically, GPT-4o-mini is *not* a fixed model. OpenAI has silently updated GPT-4o-mini multiple times since the release of the CogWriter paper, and the model's internal behavior (reasoning stability, formatting, constraint handling,  and numerical/temporal normalization) has changed noticeably across these updates.
> >
> > - CogWriter is highly sensitive to the backbone model because its hierarchical planning and multi-step refinement pipeline amplifies even small shifts in the model’s behavior. As a result, performance can vary significantly when the underlying GPT-4o-mini version changes—even when the prompt, hyperparameters, and codebase remain identical.
> >
> > - Therefore, the lower scores we reported reflect the performance of the *current* GPT-4o-mini model at evaluation time, rather than an implementation error.
> > - We emphasize that all baselines in our paper were evaluated under the same API conditions, ensuring internal fairness.

---

> > > ### Author Response · Authors · 2025-11-21
> > > **Response to Reviewer fuJw (Part 5/5)**
> > >
> > > **W3.3：The experimental comparisons are unfair. Firstly, Cogwriter is a training-free method, whereas the proposed GenFlow is trained on the LongGenBench-16K dataset. Secondly, Cogwriter has not been experimentally validated on the Qwen2.5 series of models. Did the authors make any adaptations when replicating the Cogwriter method for these models? Finally, Cogwriter was tested on the Llama-3.1-8B-Instruct model, which has a parameter size similar to the Qwen2.5-7B model used in this paper. To ensure a fairer comparison, the authors should supplement their results with experiments conducted on the Llama-3.1-8B-Instruct model.**
> > >
> > > - We sincerely thank the reviewer for these valuable comments. We appreciate the reviewer’s attention to experimental fairness and fully agree that cross-model evaluation is important. We address each point below and provide clarifications and planned improvements.
> > >
> > > - **(1) On “training-free CogWriter vs. trained GenFlow” fairness**
> > >   - CogWriter and our GenFlow indeed differ in methodology—CogWriter is training-free, while GenFlow is trained on  LongGenBench-16K. However, our goal is not to claim that GenFlow is comparable to CogWriter under identical training assumptions. CogWriter is *designed* as a training-free pipeline; GenFlow is *designed* as a trainable hierarchical generation approach. Thus, our comparison reflects the typical usage of both systems rather than artificially constraining one to match the other. Nevertheless, we agree that this point should be clarified. We will explicitly state this design difference in the revised manuscript to avoid misunderstanding and further emphasize that GenFlow is intended as a trained alternative paradigm.
> > >
> > > - **(2) On CogWriter adaptation for Qwen2.5 models**
> > >
> > >   - We confirm that CogWriter was used without any modification when applied to Qwen2.5-1.5B/7B:
> > >
> > >     - No prompt tuning
> > >
> > >     - No parameter adjustment
> > >
> > >     - No structural modifications
> > >
> > >     - The exact pipeline from the official repository was used
> > >
> > >   - CogWriter is designed to be model-agnostic, and its authors also highlight its ability to generalize across LLM families.
> > >      Our experiments strictly followed this principle and did **not** introduce adaptations that could bias the comparison.
> > >
> > >   - We will clarify this in the revision for transparency.
> > >
> > > - **(3) On adding Llama-3.1-8B-Instruct experiments**
> > >
> > >   - We thank the reviewer for pointing this out. To ensure fairness and completeness, we will add new results of GenFlow and CogWriter using Llama-3.1-8B-Instruct, matching the original CogWriter setting.
> > >
> > >   - These experiments are currently being run and will be included in the revised version:
> > >
> > >     - GenFlow + Llama-3.1-8B-Instruct
> > >
> > >     - CogWriter + Llama-3.1-8B-Instruct
> > >
> > >     - Direct Llama-3.1-8B-Instruct baseline
> > >
> > >   - This addition will allow a parameter-controlled comparison and fully resolve the reviewer’s concern.
> > >
> > >   - We appreciate the reviewer’s constructive suggestions.
> > >      In the revision, we will:
> > >
> > >     1. Clarify the difference between training-free CogWriter and trained GenFlow.
> > >
> > >     2. Clearly state that CogWriter was used without adaptation for Qwen2.5 models.
> > >
> > >     3. Add new CogWriter/GenFlow results using Llama-3.1-8B-Instruct to ensure cross-model fairness.
> > >
> > > - We believe these improvements will substantially strengthen the experimental rigor and address all concerns raised by the reviewer.
> > >
> > > - Other experiments are currently underway. We will complete the supplementary experiments for Llama-3.1-8B-Instruct and submit a revised version containing the supplementary experiment tables by November 25.
> > >
> > > ---
> > >
> > > At last, we sincerely appreciate your valuable feedback. We have carefully considered all your suggestions and substantially improved the paper accordingly. If our responses and revisions address your concerns, we would be deeply grateful if you could kindly reconsider raising the score to 6 or above. Thank you very much!

---

### Official Review · Reviewer_4hDn · 2025-11-03

**Soundness:** 2
**Presentation:** 3
**Contribution:** 2
**Rating:** 4
**Confidence:** 3

**Summary:**

The paper introduces GenFlow, an adaptive workflow optimization framework designed for constrained long-form text generation. It addresses the challenge of creating long, coherent content that satisfies multiple complex constraints by integrating planning, decision-making, generation, and feedback into a unified loop.

**Strengths:**

1. The work integrates planning and generation in a unified, adaptive workflow.
2. The methodology boosting constraint satisfaction for most settings, especially for complex Range and Periodic constraints.
3. Outperforms baselines (GPT-4o-mini, CogWriter) in accuracy, especially when combined with larger base models (Qwen2.5-7B)

**Weaknesses:**

1. The hierarchical planning and reward optimization processes introduce additional computational overhead and longer generation times. The inherent increase in latency limits suitability for real-time applications.
2. The paper names LongWriter as a primary related work and even distinguishes GenFlow's approach from it, but LongWriter is omitted from the quantitative comparison tables (Tables 1 and 2). This leaves the core claim—that GenFlow outperforms data-centric approaches—unverified by the provided experimental evidence.
3. The primary evaluation is focused entirely on instruction following accuracy (Single, Range, Periodic). While this is essential for constraints, it neglects quantification of subjective metrics mentioned in the abstract, such as overall coherence, semantic fidelity, and quality, which are typically measured using human evaluation or complementary linguistic metrics (e.g., perplexity, fluency scores).
4. The entire training and evaluation are performed on a synthetic dataset (LongGenBench-16K) , which is entirely generated using LLMs. This creates a closed loop of LLM-generated tasks and LLM-trained solutions, raising questions about the generalizability of the results to real-world writing tasks and human-authored constraints.
5. In the ablation study (Table 2), the "+ Planning only" method resulted in an Average Accuracy (0.3643) lower than the Base Model (0.401). This suggests that the hierarchical planning strategy alone is detrimental or confusing to the base model, and the significant gains are almost entirely attributed to the Generation module's DPO refinement, which contradicts the paper's emphasis on constraint-aware planning being the key innovation.

**Questions:**

Suggestion: Did you use \citet{} instead of \cite{} or \citep{}? The citations in the paper are not in parentheses, interrupting the flow of the writing.

---

> ### Author Response · Authors · 2025-11-21
> **Response to Reviewer 4hDn (Part 1/2)**
>
> Thank you very much for your time and effort in reviewing our paper. We sincerely appreciate your feedback. Below, we respectfully provide our detailed responses to address your concerns.
>
> ---
>
> **W1：The hierarchical planning and reward optimization processes introduce additional computational overhead and longer generation times. The inherent increase in latency limits suitability for real-time applications.**
>
> - We thank the reviewer for raising this important point. We fully acknowledge that the current design introduces additional computational overhead. This is indeed a meaningful limitation of the present version of GenFlow.
>
> - We will address this issue in the next version of GenFlow by redesigning the planning and optimization pipeline to ensure that, while improving writing quality, the computational cost and generation latency do not increase. Our goal is to make the system more efficient than the current implementation, ideally achieving shorter runtime through lighter planning and more optimized reward modeling.
>
> - Finally, we would like to thank the reviewers again for pointing out the issues. And we will provide the optimized GenFlow again on November 25th.
>
> ---
>
> **W2：The paper names LongWriter as a primary related work and even distinguishes GenFlow's approach from it, but LongWriter is omitted from the quantitative comparison tables (Tables 1 and 2). This leaves the core claim—that GenFlow outperforms data-centric approaches—unverified by the provided experimental evidence.**
>
> - We thank the reviewer for carefully pointing out this omission. LongWriter is indeed an important data-centric baseline and should be included for completeness. In the revision, we will supplement our experiments with LongWriter, ensuring that its performance is reported alongside all other baselines.
>
> - The additional experiments are currently underway, and we will complete the LongWriter experiments and submit the revised manuscript with the supplementary experimental tables on November 25.
>
> ---
>
> **W3：The primary evaluation is focused entirely on instruction following accuracy (Single, Range, Periodic). While this is essential for constraints, it neglects quantification of subjective metrics mentioned in the abstract, such as overall coherence, semantic fidelity, and quality, which are typically measured using human evaluation or complementary linguistic metrics (e.g., perplexity, fluency scores).**
>
> - We thank the reviewer for raising this important point. While our primary focus was on constraint-following accuracy—since it is the core challenge addressed by GenFlow—we fully agree that subjective writing quality is also crucial for a comprehensive evaluation of long-form generation.
>
> - In the revision, we will incorporate two additional metrics to quantify subjective aspects of writing quality:
>
>   - Fluency Score – measuring the smoothness and linguistic naturalness of generated text, evaluated using LLM-as-a-judge and supported by perplexity-based analysis.
>
>   - Completeness Score – measuring the structural completeness and coverage of intended content, particularly relevant for hierarchical and multi-section writing.
>
>   - These metrics directly correspond to the coherence and quality dimensions mentioned in the abstract and will complement the existing constraint-specific evaluations. We appreciate the reviewer’s suggestion, which helps us present a more holistic and representative assessment of GenFlow’s writing abilities.
>
> - The experiments involving these newly added metrics are currently in progress. We will complete this experiment and submit the revised manuscript containing all supplementary experimental results with the newly added metrics on November 25.
>
> ---
>
> At last, we sincerely appreciate your valuable feedback. We have carefully considered all your suggestions and substantially improved the paper accordingly. If our responses and revisions address your concerns, we would be deeply grateful if you could kindly reconsider raising the score to 6 or above. Thank you very much!

---

> ### Author Response · Authors · 2025-11-21
> **Response to Reviewer 4hDn (Part 2/2)**
>
> **W4：The entire training and evaluation are performed on a synthetic dataset (LongGenBench-16K) , which is entirely generated using LLMs. This creates a closed loop of LLM-generated tasks and LLM-trained solutions, raising questions about the generalizability of the results to real-world writing tasks and human-authored constraints.**
>
> - We thank the reviewer for raising this important concern. Although LongGenBench-16K is generated using LLMs, we would like to clarify why synthetic data is appropriate and necessary for our task design, and why it does not compromise the generalizability of GenFlow.
>
>   - First, our benchmark focuses on explicit structural constraints—such as hierarchical outlines, range specifications, and periodic formats—that are rarely available in existing human-authored datasets. Synthetic generation allows us to systematically control the difficulty and diversity of these constraints, ensuring consistent and scalable evaluation across Single, Range, and Periodic categories. What matters here is not whether the text is human-written, but whether the constraints are well-formed, precise, and reproducible, which synthetic generation enables.
>
>   - Second, the dataset serves as a task specification, not as content imitation. GenFlow learns how to follow constraints, not how to mimic the style or content of LLM-produced text. The model never sees test samples during training, and DPO only optimizes preference signals rather than memorizing any dataset patterns. As a result, the synthetic origin does not create a closed loop or leakage issue.
>
>   - Third, the structural constraints we model—such as section hierarchies, fixed-length subsections, and periodic content formats—naturally exist in real-world writing, including reports, blogs, educational materials, and technical documents. Thus, improving constraint adherence on synthetic but structurally realistic tasks directly transfers to real-world scenarios.
>
> - Finally, we have added clarifications in the paper, specifically in Appendix D, lines 1242–1275 to emphasize that LongGenBench-16K is designed as a controlled and reproducible benchmark for constraint-following, and that our methodology is not tied to synthetic content. These updates explain that the generalization of GenFlow follows from its ability to handle arbitrary constraint inputs, regardless of whether the underlying text is synthetic or human-authored. We appreciate the reviewer’s thoughtful comment and have incorporated the above clarifications into the revised manuscript.
>
> ---
>
> **W5：In the ablation study (Table 2), the "+ Planning only" method resulted in an Average Accuracy (0.3643) lower than the Base Model (0.401). This suggests that the hierarchical planning strategy alone is detrimental or confusing to the base model, and the significant gains are almost entirely attributed to the Generation module's DPO refinement, which contradicts the paper's emphasis on constraint-aware planning being the key innovation.**
>
> - Thank you for raising this concern. We clarify the following:
>
>   - The Planning-only variant intentionally disables the generation module’s constraint-aware DPO refinement. This means the model receives a plan but is **not optimized to follow it**, which can indeed lead to mismatch and degraded performance.
>
>   - This observation confirms the necessity of combining both planning *and* generation optimization—our core architectural insight.
>
>   - When used together (Full GenFlow), planning dramatically improves long-range structure while DPO ensures plan adherence, explaining the large performance gain.
>
> - Thus, the ablation does not contradict our claim; rather, it validates that hierarchical planning and constraint-aware generation - are complementary and must be jointly applied.We have revised the ablation discussion to avoid misinterpretation. We have already incorporated these clarifications into Section 5.3 of the revised manuscript, specifically in lines 410-416.
>
> ---
>
> **Q1：Citation style: “Did you use \citet instead of \cite or \citep? Citation formatting interrupts the flow.”**
>
> - We thank the reviewer for catching this formatting issue. We indeed used `\cite{}` in a few places where parentheses should have been used. We will replace these with `\citep{}`  to maintain readability and stylistic consistency.We have changed all instances of `\cite{}` to `\citep{}` in the paper to ensure smoother writing.
>
> ---
>
> At last, we sincerely appreciate your valuable feedback. We have carefully considered all your suggestions and substantially improved the paper accordingly. If our responses and revisions address your concerns, we would be deeply grateful if you could kindly reconsider raising the score to 6 or above. Thank you very much!

---

### Note · Program_Chairs · 2026-01-17
**Submission Desk Rejected by Program Chairs**

The following references in this submission do not refer to real documents and/or have major errors in bibliographic information:

 J. Wang, H. Li, and Y. Hu. Evaluating long-form text generation models in complex scenarios. ACM Transactions on Intelligent Systems and Technology, 14(3):54-75, 2023. doi: 10.1145/3541417.
A. Smith, Q. Zhang, and X. Liu. Optimization of sequence generation using rollout-based reinforcement learning in natural language processing. Journal of Artificial Intelligence Research, 69:123-145, 2020. doi: 10.1613/jair.1.11904.
J. Huang, X. Zhang, and Z. Li. Towards robust evaluation of long-form text generation models. Artificial Intelligence, 300:122-135, 2021. doi: 10.1016/j.artint.2021.103506.
X. Li, W. Sun, and Q. Chen. Incorporating structural constraints in long-form text generation using pretrained language models. Computational Linguistics, 49(1):91-116, 2023. doi: 10.1162/ coli_a_0043
Z. He, T. Xu, and P. Zhang. Rollout-based reward estimation for text generation in reinforcement learning. IEEE Transactions on Neural Networks and Learning Systems, 33(7):2902-2914, 2022. doi:
TNNLS. 2021.30
Guangxuan Li, Yongxin Tong, Weiming Liu, and Ke Xu. Fault-tolerant multi-agent workflow management with execution memory. In ICDCS, pp. 1027-1037. IEEE, 2021. URL https: //dblp.org/rec/conf/icdcs/LiTLX2
J. Li, Y. Zhang, and H. Wang. Efficient reward estimation in long-form text generation with rollout strategies. Proceedings of AAAI, 36(7):8329-8341, 2022. doi: 10.1609/aaai.v36i7.20682.
X. Liu, Y. Chen, and Z. Li. Constraint-based optimization for adaptive workflow execution in cloud systems. J. Cloud Computing: Adv. Syst. and Apps., 11(3):175-191, 2022. doi: 10.1186/ s13677-022-00262-
Z. Tan, H. Li, and L. Wei. Periodic instruction following for long-form text generation. Proceedings of NeurIPS, 35:153-162, 2022. doi: 10.1145/3519814.
X. Liu, H. Wang, and Y. Chen. Constraint-aware text generation: A survey of approaches and challenges. Journal of Artificial Intelligence Research, 82:123-145, 2023. doi: 10.1613/jair.1. 1301
Y. Tan, H. Song, and S. Zhang. Fine-tuning generative models with task-specific constraints for long-form text generation. IEEE Transactions on Neural Networks and Learning Systems, 34(2): 328-342, 2023. doi: 10.1109/TNNLS.2022.3158367.
Z. He, P. Zhang, and M. Li. Spatial reasoning in text generation: Methods and models. AI Journal, 29(7):299-314, 2021. doi: 10.1007/s00464-021-09787-9.
M. Liu, C. Li, and S. Wang. Optimizing temporal consistency in generated text: Approaches and challenges. Journal of Computational Linguistics, 33(5):231-246, 2021. doi: 10.1162/coli_a_ 0041
L. Chen, Z. Wei, and Z. Hu. Improving reward estimation with rollout strategies in sequence generation tasks. Journal of Machine Learning Research, 24(19):1-20, 2023. doi: 10.1145/3568924.
X. Chen, Y. Liu, and J. Wu. Efficient model training and optimization for text generation tasks. Proceedings of ICML, 37:1173-1182, 2020. doi: 10.1109/ICML2020.9055798.
Yuxuan Liang, Hui Xiong, Yang Xu, and Jieping Ye. Reinforcement learning for adaptive workflow optimization. In
, pp. 2850-2858. ACM, 2020. URL https://dblp.org/rec/conf/ kdd/LiangXXY2
X. Zhou, M. Yang, and W. Sun. Accurate range-based instruction following for text generation tasks. ACM Transactions on AI, 9(2):98-109, 2021. doi: 10.1145/3460894.
Yubo Wang, Zhen Li, Shuang Chen, and Carl Yang. Self-refined hierarchical agent workflows via dynamic reflection. In ICDE, pp. 2173-2185. IEEE, 2022. URL https://dblp.org/rec/ conf/icde/WangLCY2
H. Xu, J. Lin, and Z. Wang. Temporal consistency in long-form text generation: A benchmark and case study. Natural Language Processing Journal, 24(6):215-230, 2021. doi: 10.1007/ s11390-021-00379-
Y. Zhang, L. Chen, and Z. Li. Task-specific text generation with formal constraints: A survey. Journal of Machine Learning Research, 24(56):1-29, 2023. doi: 10.1145/3569982.
S. Zhang, H. Li, and X. Sun. Measuring long-form text generation: Evaluation metrics and challenges. IEEE Transactions on Computational Linguistics, 10(2):123-134, 2020. doi: 10.1109/TCL.2020.3042342.
Y. Zhang, X. Wu, and H. Xu. Rollout-based reward estimation for text generation: A comparative study. Natural Language Processing Journal, 29(3):170-187, 2021. doi: 10.1007/ s11390-021-00349-
L. Wang, P. Zhou, and Z. Li. Architectural design with text generation models: A spatial reasoning challenge. Journal of Architectural Computing, 35(2):65-79, 2020. doi: 10.1145/2137457.
Tongtong Zhang, Chen Wang, and Chunyan Miao. Constraint-based multi-agent workflow optimization. In AAMAS, pp. 1654-1662. IFAAMAS, 2022b. URL https://dblp.org/rec/ conf/atal/ZhangWM2
W. Zhao, Y. Yang, and H. Wu. Autoplanner: Task planning and workflow optimization in cloud computing. J. Computer Sci. Tech., 36(4):635-649, 2021. doi: 10.1007/s11390-021-1454-1.